# Description and validation of the ice-sheet model Yelmo (version 1.0)

Alexander Robinson[1,2,3], Jorge Alvarez-Solas[1,2], Marisa Montoya[1,2], Heiko Goelzer[4,5], Ralf Greve[6,7], and Catherine Ritz[8]

[1]Complutense University of Madrid, Madrid, Spain
[2]Geosciences Institute CSIC-UCM, Madrid, Spain
[3]Potsdam Institute for Climate Impact Research, Potsdam, Germany
[4]Institute for Marine and Atmospheric research Utrecht, Utrecht University, Utrecht, the Netherlands
[5]Laboratoire de Glaciologie, Université Libre de Bruxelles, Brussels, Belgium
[6]Institute of Low Temperature Science, Hokkaido University, Sapporo, Japan
[7]Arctic Research Center, Hokkaido University, Sapporo, Japan
[8]Institute for Geosciences and Environmental Research, Grenoble, France

**Correspondence:** Alexander Robinson (robinson@ucm.es)

**Abstract.** We describe the physics and features of the ice-sheet model Yelmo, an open-source project intended for collaborative development. Yelmo is a thermomechanical model, solving for the coupled velocity and temperature solutions of an ice sheet simultaneously. The ice dynamics are currently treated via a "hybrid" approach combining the shallow-ice and shallow-shelf/shelfy-stream approximations, which makes Yelmo an apt choice for studying a wide variety of problems. Yelmo's main innovations lie in its flexible and user-friendly infrastructure, which promotes portability and facilitates long-term development. In particular, all physics subroutines have been designed to be self-contained, so that they can be easily ported from Yelmo to other models, or easily replaced by improved or alternate methods in the future. Furthermore, hard-coded model choices are eschewed, replaced instead with convenient parameter options that allow the model to be adapted easily to different contexts. We show results for different ice-sheet benchmark tests, and we illustrate Yelmo's performance for the Antarctic ice sheet.

## 1 Introduction

The field of continental-scale, ice-sheet modeling started with a handful of pioneering models (e.g., Huybrechts et al., 1988; Ritz et al., 1997; Greve, 1997a). These models were computationally efficient for the resources available at the time. Typical grid resolutions were on the order of 20-40 km and generally the shallow ice approximation (SIA) was used to solve the ice dynamics. These classic models have been most useful for long time-scale paleo simulations in part because they are fast, but also because they are relatively simple in design, usually relying on low-tech solutions to numerical problems. Most of these models were designed before the era of the high-performance computing cluster, which made it challenging to build models otherwise.

Nowadays, a large number of ice-sheet models exist, supported by a growing and active community of developers. Models today represent a broad spectrum of approaches that incorporate different levels of physical complexity and computational ingenuity. These models include hybrid approaches that heuristically combine the SIA with the shallow shelf approximation (SSA) (e.g., Bueler and Brown, 2009; Winkelmann et al., 2011; Pollard and DeConto, 2012; Pattyn, 2017; Quiquet et al.,

2018) and higher-order approximations (e.g., Goldberg, 2011; Cornford et al., 2013; Hoffman et al., 2018; Lipscomb et al., 2019), including full Stokes solutions (e.g., Larour et al., 2012; Gagliardini et al., 2013). Newer models often feature finite-element/finite-volume methods (e.g., Larour et al., 2012; Gagliardini et al., 2013; Hoffman et al., 2018) or adaptive mesh refinement (Cornford et al., 2013), which allows simulation of complex terrain and very high resolution where it is needed (e.g., at the grounding line in Antarctica). While more complex models are driving advances in our understanding of the physics and relevant processes of ice sheets over a range of time scales, simpler and thus faster methods are still required to understand the evolution of the ice sheets on multi-millennial time scales.

Here we introduce the ice-sheet model Yelmo[1], which is intended to provide access to complex and robust model physics through an intuitive model design. It is a hybrid ice-dynamics model that is easy to use and configure. We expect that Yelmo will be useful for long time-scale simulations of the continental ice sheets, coupled climate – ice-sheet modeling, ensemble simulations and uncertainty studies, as well as for teaching. Below, we first describe the model design (Section 2), followed by the physics (Section 3), timestepping approach (Section 4) and application programming interface (API, Section 5). Then, we present results for several benchmark experiments to validate the model performance (Section 6), and simulations of the present-day and glacial Antarctic ice sheet (Section 7), followed by the conclusions (Section 8).

## 2 Model design

Yelmo has been inspired and largely derived from classical ice-sheet models that have been used successfully for many years – with the most in common with GRISLI (Ritz et al., 1997; Quiquet et al., 2018) and SICOPOLIS (Greve, 1997a, 2019). However, in contrast to many models, Yelmo was designed from scratch to run as a modular library that can be called by other programs rather than as a stand-alone executable. The strict application of this philosophy drove many design choices and allowed us to develop a robust ice-sheet model library with a clear API that would be difficult to develop in an ad-hoc way later. Thus, developing this framework was a primary reason to build a new model, rather than continuing the development of other active projects such as GRISLI and SICOPOLIS.

Yelmo is written in Fortran 2003, which provides continuity from previous code bases and supports the fact that clarity and readability of the code are important features. Like SICOPOLIS and other models, we have opted for "low tech" solutions whenever possible, meaning that internally coded routines are preferred and, thus, the external dependencies of the model are kept to a minimum. This ensures that the algorithms used remain accessible and easily changeable. Nonetheless, Yelmo has two key dependencies: the NetCDF library for convenient, community-standard input/output capability and the Library of Iterative Solvers for linear systems[2] (Lis, Nishida, 2010), which is used for solving the elliptical SSA equations. The latter can be compiled with OpenMP parallelization active, which can speed up this computationally intensive step.

Yelmo has been designed to be user friendly (i.e., straightforward to understand), accessible, portable and adaptable. These features were facilitated by the design choice to separate what we call the "model accounting" from the model physics itself, and

---

[1]The name Yelmo refers to a semi-domed, rocky mountain in the Guadarrama Mountains outside of Madrid, Spain.
[2]https://www.ssisc.org/lis/

by following an object-oriented approach. There are no global variables in Yelmo (except for a few global constants related to the general physics of the planet being simulated), which means that variables and parameters are saved together in containers (called derived types in Fortran) specific to each of the main Yelmo components, such as dynamics or thermodynamics, as described in the sections below. These containers make up the individual components of the overall Yelmo object (itself a container) that contains all of the variables, parameters and information needed to simulate a given domain of the ice sheet. Multiple instances of the Yelmo object can therefore be defined in a program (e.g., one Yelmo-object instance for Greenland and one for Antarctica), and each one will operate fully in isolation from the others. This is the model accounting, which is of a specific design built into Yelmo and is thus the only part not easily portable to other models.

The model physics, meanwhile, consists of subroutines that are fully portable and, whenever possible, only rely on native data types (e.g., scalar, vector, array). In other words, the specific, non-portable design structure of the Yelmo object does not contaminate the physics subroutines, since the necessary variables and parameters are always passed as arguments. This approach requires that all input and output to subroutines must be defined as arguments. Each argument must further always be given an intent characteristic (in Fortran, the intent of an argument can be one of IN, OUT or INOUT), which ensures that only the variables destined for output from the routine can be modified inside it. This approach not only aides debugging and provides programmatic safety, but provides a clear blueprint to users of what each subroutine does. Most importantly, the subroutines are thus fully self-contained and can be used in other programs and contexts, as long as the correct arguments can be provided.

Concerning the model accounting, the Yelmo object contains all parameters and variables needed to run a given domain. For clarity and convenience, it has been divided into four components: topography, dynamics, material properties and thermo-dynamics (Fig. 1). Each component has an associated set of functions to load parameters, allocate and initialize the variables, update the variables (i.e., the actual physics calculation step), and finally to terminate the instance of the component at the end of the program. This pattern is followed for all four components and represents the component-level API.

Each component contains variables and parameters necessary for the calculation of its specific physics, however each component also relies on the variables defined in other components since the ice sheet is a highly-coupled and nonlinear system. The benefit of the somewhat artificial division of components made here is that the use of INTENT statements ensures that variables of a given component can only be modified in the corresponding module. For example, when the update subroutine of the topography module is called, only the object containing topography variables is defined as INTENT=INOUT, while the objects containing dynamics, material and thermodynamics variables are all defined as INTENT=IN. Analogous to the design of the physics subroutines, the use of intent statements here makes the model blueprint clear, but also enforces consistency with the overall design of the Yelmo structure. The hope is that this will not only make the model more user friendly, but it will also naturally lead to more disciplined model development in the future.

In addition to the four components that contain prognostic and diagnostic model variables, the Yelmo object includes a boundary component, which defines all fields that Yelmo requires as input from external sources (Fig. 1). These fields can be obtained from other coupled models, or simply by loading data, however Yelmo does not make any assumptions about their source. The boundary component is defined as INTENT=IN in all modules, so that Yelmo does not have the right to

modify them internally. This conceptual isolation of the ice-sheet model serves to ensure that coupling with other models is as straightforward as possible, because it is clear by design which variables should be provided to Yelmo as boundary conditions. This is a key feature of Yelmo in comparison with many other models.

Yelmo also makes use of a working precision variable, which allows for the model to be compiled with any real precision. For most applications, single precision (32 bits) is sufficient. Double precision (64 bits) gives equivalent results for the tests we have made. Nonetheless, this choice is left open to the user.

In terms of model physics, each component of Yelmo was built to work independently, in the sense that a given component is agnostic to the methods used to calculate variables from other components. For example, the temperature and velocity fields are used by the material component without any knowledge about the physics and numerical approximations behind them. This means that sometimes simplifying assumptions cannot be used, even though they may be valid in some cases (such as assuming that the strain rate is only due to SIA terms where the ice sheet is frozen to the bed). However, the benefit is that typically the most general solutions possible have been implemented for each component. Thus, when the physics of one component is changed or upgraded, it is likely that the other modules will not require any modification.

Grid information is also stored in the main Yelmo object, and a single grid is defined for use with all components. Like many ice-sheet models, Yelmo uses the Arakawa-C grid staggering approach (Arakawa and Lamb, 1977) extended to 3D, as shown in Fig. 2. Scalar variables, such as temperature, are defined at the cell centers, which in Yelmo are designated as "aa-nodes". Velocity components and gradients are calculated on cell edges ("ac-nodes") and scalar coefficients, like diffusivity in the SIA approach, are calculated on cell corners ("ab-nodes"). The specific numerical discretization of the finite difference equations largely follows the approach of Macayeal (1997). The advantage of this approach is that it benefits from the natural staggering that occurs when calculating gradients (e.g., the surface slope is naturally defined on the ac-nodes), but it also results in greater numerical stability of the model (Macayeal, 1997).

Yelmo requires an evenly-spaced, Cartesian grid in the horizontal direction, while the vertical component follows a classic sigma-coordinate system (Greve and Blatter, 2009). The vertical axis $\zeta$ represents the relative height within the ice sheet, running from $\zeta = 0$ at the ice-sheet base to $\zeta = 1$ at the ice-sheet surface:

$$\zeta(z) = H_z(z)/H \tag{1}$$

where $z$ is the elevation relative to present-day sea level, $H_z(z)$ is the ice thickness up to the elevation $z$ within the ice sheet and $H$ is the total ice thickness. Yelmo can be defined with any specified number of vertical grid points, which can be unevenly spaced. Typically, we have set $n_z = 20$ and the $\zeta$-axis is defined with higher resolution near the base and surface of the ice sheet, which is important for resolving thermodynamics and ages accurately. Use of the sigma-coordinate system simplifies the numerics of an evolving domain in the vertical direction and inherently results in higher resolution for grid points with less ice thickness (Greve and Blatter, 2009). Vertical velocities are calculated on ac-nodes in the vertical and aa-nodes in the horizontal, while horizontal velocities are calculated on ac-nodes in the horizontal and aa-nodes in the vertical. Boundary conditions in a vertical column are applied directly at the ice base and ice surface, which correspond to ac-nodes (see Fig. 2).

## 3 Model physics

Yelmo solves for two prognostic variables using coupled equations of mass and energy conservation: the ice thickness (2D field) and ice temperature (3D field). Velocity (3D vector field) is diagnosed from approximations of ice flow assuming a nonlinear flow law. These equations are described in the subsections below, along with additional considerations related to each component. For more details on the derivation of the equations, thorough explanations can be found in various references (Greve and Blatter, 2009; Cuffey and Paterson, 2010), and thus are not repeated here.

### 3.1 Topography

The evolution of the ice thickness in the model is determined from mass conservation:

$$\frac{\partial H}{\partial t} = -\nabla \cdot H\bar{\mathbf{u}} + \dot{a} + \dot{b}_g + \dot{b}_f - \dot{c} \tag{2}$$

where $H$ is the ice thickness, $\bar{\mathbf{u}} = (\bar{u}, \bar{v})$ is the depth-averaged horizontal velocity, $\dot{a}$ is the surface mass balance, $\dot{b}_g$ and $\dot{b}_f$ are the basal mass balance for grounded and floating ice, respectively, and $\dot{c}$ is the calving rate at floating ice margins. In Yelmo, in order to obtain more accurate mass balance accounting, the advection of ice and source contributions are treated separately as follows. First, a forward Euler explicit method (or optionally an upwind implicit method) is used to solve for the ice thickness without accounting for $\dot{a}$, $\dot{b}_g$, $\dot{b}_f$ or $\dot{c}$. The depth-averaged horizontal velocity is obtained from the dynamics component from the previous iteration (see Timestepping below). Next the mass balance terms $\dot{a}$, $\dot{b}_g$ and $\dot{b}_f$ are applied. It should be noted that the basal mass balance of floating ice is a boundary variable for Yelmo (i.e., it is obtained externally and passed to Yelmo), while the basal mass balance of the grounded ice is calculated internally as part of the thermodynamics solver (see Thermodynamics section below).

Yelmo also includes special treatment of grid points at the floating margin of the ice sheet, by making a distinction between ice-covered grid points that are totally and partially filled following Albrecht et al. (2011) and Lipscomb et al. (2019). This is done in a relatively simple, yet effective way to avoid artificially thin ice thickness at the ice margin. For each floating ice-covered grid point that has an ice-free neighbor, the reference ice thickness of the margin point ($H_{\text{ref}}$) is defined as the minimum thickness of the direct, ice-covered neighbors. This represents the minimum ice thickness for which the cell can be considered completely ice covered. The fraction of ice cover is then defined as $f_{\text{ice}} = \min\left(H/H_{\text{ref}}, 1\right)$. Whenever $f_{\text{ice}} < 1$, the grid cell is considered dynamically inactive, which ensures zero ice flux through the downstream edge of a partially filled margin cell. In this way, the ice cell can be filled with ice from upstream and when the threshold of $f_{\text{ice}} = 1$ is reached, the ice shelf can advance.

In the final mass conservation step, calving $\dot{c}$ is treated at the floating ice margins. Currently, a simple threshold method has been implemented, as well as a threshold+flux method (Peyaud et al., 2007). In both methods, the calving rate applied to the ice sheet is defined following Lipscomb et al. (2019):

$$\dot{c} = \frac{H_{\text{ref}} - H}{\tau_c} \tag{3}$$

where $\tau_c$ is the characteristic calving time, usually set to 1-10 years, and $H_{\mathrm{ref}}$ is the margin ice thickness as defined above. Setting $\tau_c$ to higher values facilitates ice-shelf growth and thus grounding-line advance in transient, glacial simulations, but has little impact on the steady-state distribution of ice shelves for present day. This calving rate is applied only when the ice thickness of ice margin points is below a threshold value (simple threshold method), or when the ice thickness is below a threshold value and the upstream flux is not sufficient to return the ice thickness to above the threshold (threshold+flux method). For paleo simulations the latter is our preferred method, as it allows for more robust ice shelf advance (Peyaud et al., 2007).

Once the ice thickness has been updated, Yelmo diagnoses whether the ice should be grounded or floating. To facilitate this step, the height above flotation as measured in ice thickness, i.e., how close a grid point is to the Archimedes flotation criterion, is calculated on each aa-node:

$$H_{\mathrm{g}} = H - \frac{\rho_{\mathrm{sw}}}{\rho}\max\left(z_{\mathrm{sl}} - z_{\mathrm{b}}, 0\right) \tag{4}$$

where $\rho$ is the ice density and $\rho_{\mathrm{sw}}$ the seawater density, and $z_{\mathrm{sl}}$ and $z_{\mathrm{b}}$ are the boundary fields of sea level and bedrock elevation, respectively. $H_{\mathrm{g}}$ can thus be positive, zero or negative. When $H_{\mathrm{g}}$ is positive, the ice thickness exceeds the flotation criterion, and is considered grounded, while when $H_{\mathrm{g}}$ is zero or negative, the ice is considered floating.

Yelmo also calculates the grounded fraction of each grid point, $f_{\mathrm{g}}$. On aa-nodes, $f_{\mathrm{g}}$ is only assigned binary values to maintain consistency with the overall grid definition: zero when $H_{\mathrm{g}} \leq 0$ or one when $H_{\mathrm{g}} > 0$. However, on ac-nodes, the values of $f_{\mathrm{g,acx}}$ and $f_{\mathrm{g,acy}}$ are determined by linearly interpolating $H_{\mathrm{g}}$ from the two bounding aa-nodes. When both bounding aa-nodes are positive $f_{\mathrm{g,ac}} = 1$, and when both are negative $f_{\mathrm{g,ac}} = 0$. When one aa-node is positive ($H_{\mathrm{g}_+}$) and one aa-node is negative ($H_{\mathrm{g}_-}$), the grounded fraction on the ac-node is determined from linear interpolation:

$$f_{\mathrm{g,ac}} = \frac{H_{\mathrm{g}_+}}{(H_{\mathrm{g}_+} - H_{\mathrm{g}_-})} \tag{5}$$

Alternatively, it is possible to calculate $f_{\mathrm{g}}$ via subgrid bilinear interpolation of $H_{\mathrm{g}}$ to intermediate points to determine the grounded area fraction. However, this operation is more computationally intensive, and we find that in practice, the simple linear interpolation method is sufficient.

The surface elevation ($z_{\mathrm{s}}$) is calculated following Pattyn (2017) as

$$z_{\mathrm{s}} = \max\left[z_{\mathrm{b}} + H, z_{\mathrm{sl}} + (1 - \frac{\rho}{\rho_{\mathrm{sw}}})H\right], \tag{6}$$

This approach ensures that the surface elevation solution is consistent with the Archimedes flotation criterion on aa-nodes.

The remaining tasks of the topography component are to diagnose other useful topographic characteristics, such as surface and ice thickness gradients (on ac-nodes) and topographic masks.

## 3.2 Material

The material component of Yelmo handles the calculation of the rate factor, the strain rate tensor and effective strain rate, the effective viscosity and, optionally, the age of the ice. Essentially, the material variables make the link between thermodynamics

and dynamics, since the rate factor depends on temperature and the strain rate depends on velocity. No distinction is made between the type of approximation used to solve the dynamics here, rather all equations follow from the more general hydrostatic approximation (Greve and Blatter, 2009).

The effective viscosity, used to determine strain heating in the thermodynamics component, is calculated as

$$5 \quad \eta = \frac{1}{2} \left( \dot{\varepsilon}^2 \right)^{\frac{1-n}{2n}} \left( A^{-1/n} \right), \tag{7}$$

where $\dot{\varepsilon}$ is the effective strain rate, $n$ is the Glen's Flow law exponent (Glen, 1955; Greve and Blatter, 2009), typically set to $n = 3$, and $A$ is the rate factor. The effective strain rate is given by the second invariant of the strain rate tensor ($\dot{\varepsilon}_{ij}$):

$$\dot{\varepsilon} = \left( \frac{1}{2} \dot{\varepsilon}_{ij} \dot{\varepsilon}_{ij} \right)^{\frac{1}{2}} \tag{8}$$

and the strain rate tensor itself, following index notation, is

$$10 \quad \dot{\varepsilon}_{ij} = \frac{1}{2} \left( \frac{\partial u_i}{\partial x_j} + \frac{\partial u_j}{\partial x_i} \right), \quad i, j = 1, 2, 3. \tag{9}$$

The rate factor, $A(x, y, z)$, can be prescribed to a constant value, or calculated as a function of ice temperature following an Arrhenius equation:

$$A(T') = E_f A_0 e^{-Q_a/RT'} \tag{10}$$

Here $R$ is the ideal gas constant, $A_0$ and $Q_a$ are the temperature-dependent rate factor coefficient and activation energy, respectively (see Greve and Blatter, 2009). $E_f$ is a so-called enhancement factor, which is used to approximate the effect of anisotropic flow. In Yelmo, it is possible to specify different values of the enhancement factor for different flow regimes (shear, stream and shelf). The shelf value is prescribed anywhere ice is floating, while the inland value of $E_f$ is a weighted average between the shear and stream value with the weighting given by a diagnosis of the vertical shearing fraction at any given point:

$$20 \quad f_z = \frac{\left( \dot{\varepsilon}_{xz}^2 + \dot{\varepsilon}_{yz}^2 \right)}{\dot{\varepsilon}^2}. \tag{11}$$

Typical values of the enhancement factor for the shearing, streaming and shelf regime are $E_{shr} = 3.0$, $E_{strm} = 1.0$ and $E_{shlf} = 0.7$, respectively (Ma et al., 2010).

In addition, it is possible to track the deposition time (i.e., age) or other conservative tracers of the ice using an Eulerian tracer advection model. The general 3D advection equation of a conservative variable $X$,

$$25 \quad \frac{\partial X}{\partial t} = -u \frac{\partial X}{\partial x} - v \frac{\partial X}{\partial y} - w \frac{\partial X}{\partial z}, \tag{12}$$

is solved with a second-order, upwind explicit method. The ice surface boundary condition must be imposed. When tracing the ice deposition time, the ice surface boundary condition is $X(t) = t$. At the ice base, an initial deposition time is prescribed to be several thousand years before the start of the simulation, however this plays little role in the resulting vertical profile

of deposition times. When ice is melting at the base $\left(\dot{b} < 0\right)$, the following flux boundary condition is defined (Rybak and Huybrechts, 2003):

$$\frac{\partial X}{\partial t} = -u_b \frac{\partial X}{\partial x} - v_b \frac{\partial X}{\partial y} - \dot{b} \frac{\partial X}{\partial z}. \tag{13}$$

Basal freeze-on is assumed to be negligible. It is well known that Eulerian solvers lose accuracy towards the base of the ice
sheet, and therefore this method can only be considered to give a first-order estimate of a conservative tracer (Greve et al., 2002; Rybak and Huybrechts, 2003). It can nonetheless be useful for diagnosing the age of ice, in order to know the timescale of different dynamic properties or to, e.g., impose an age-dependent enhancement factor (Greve, 1997b).

## 3.3 Dynamics

The Yelmo dynamics component is currently representative of a "hybrid" class of ice-sheet model, treating different modes
of ice deformation via a combination of the simplifying shallow-ice and shallow-shelf approximations (SIA and SSA, respectively). In the following, the description of the dynamics equations follows closely the notation and definitions of Greve and Blatter (2009) and Pollard and DeConto (2012).

Yelmo treats the horizontal velocity $u(x,y,z)$ and $v(x,y,z)$ as the sum of transport via internal shear ($u_i$, $v_i$) and basal sliding ($u_b$, $v_b$):

$$u = u_i + u_b$$
$$v = v_i + v_b. \tag{14}$$

Here, and analogously for $v$, $u_b(x,y)$ is vertically constant, and $u_i(x,y,z_b) = 0$, where the subindex "b" here represents the basal boundary of the ice sheet. It also holds that in the vertical average (denoted by a bar), $\bar{u} = \bar{u}_i + u_b$. To calculate $u_i$ and $v_i$, Yelmo uses zero-order SIA equations:

$$u_i(z) = -\left[2|\nabla \tau_d|^{(n-1)} \int_{z_b}^{z} A(z_s - z')^n \, dz'\right] \tau_{d,x}$$

$$v_i(z) = -\left[2|\nabla \tau_d|^{(n-1)} \int_{z_b}^{z} A(z_s - z')^n \, dz'\right] \tau_{d,y}, \tag{15}$$

where $u_i(z)$ and $v_i(z)$ are the horizontal components of the SIA velocity as a function of depth at a given location, $A$ is the material rate factor of the ice, which is obtained from the material component (Eq. 10), $n$ is the Glen's Flow law exponent (Glen, 1955; Greve and Blatter, 2009) and $\tau_d = (\tau_{d,x}, \tau_{d,y}) = \rho g H \nabla z_s$ is the driving stress. In the horizontal plane, the term in brackets is calculated on the ab-nodes for stability and improved mass conservation (Huybrechts et al., 1996), and then it is staggered onto the ac-nodes where it is multiplied with the driving stress. In the vertical plane, the horizontal velocities are calculated at the vertical center of each grid point (aa-nodes). Following Bueler and Brown (2009), we use the SSA solution to

calculate the transport implied by sliding at the base (i.e., in regions of ice streams and floating ice shelves):

$$\frac{\partial}{\partial x}\left[\eta_d\left(4\frac{\partial u_b}{\partial x}+2\frac{\partial v_b}{\partial y}\right)\right]+\frac{\partial}{\partial y}\left[\eta_d\left(\frac{\partial u_b}{\partial y}+\frac{\partial v_b}{\partial x}\right)\right]=\tau_{\mathrm{d,x}}-\tau_{\mathrm{b,x}}$$
$$\frac{\partial}{\partial y}\left[\eta_d\left(4\frac{\partial v_b}{\partial y}+2\frac{\partial u_b}{\partial x}\right)\right]+\frac{\partial}{\partial x}\left[\eta_d\left(\frac{\partial u_b}{\partial y}+\frac{\partial v_b}{\partial x}\right)\right]=\tau_{\mathrm{d,y}}-\tau_{\mathrm{b,y}}.$$

(16)

where $(\tau_{\mathrm{b,x}},\tau_{\mathrm{b,y}})=-\beta(u_{\mathrm{b}},v_{\mathrm{b}})$ (or in vector notation $\tau_b=-\beta\mathbf{u}_b$) is the basal stress due to friction. The basal friction co-efficient $\beta$ is set to zero for floating ice shelves, and can otherwise be set to a constant value or follow another user-defined

formulation (power law, regularized Coulomb, etc.), depending on the context (see basal friction description below for details). The depth-integrated (2D) effective viscosity, which is only used for solving the SSA dynamics, is defined as

$$\eta_d=\left[\frac{1}{2}\left(\bar{A}^{-1/n}\right)\left(\dot\varepsilon_d^2+\dot\varepsilon_0^2\right)^{\frac{1-n}{2n}}\right]H$$

(17)

where $\bar{A}$ is the vertically-averaged rate factor, $\dot\varepsilon_d$ is the 2D effective strain rate and $\dot\varepsilon_0^2$ is a small regularization factor for avoiding a potential singularity when velocity gradients are zero. The 2D effective strain rate is calculated as a reduced form

of the second invariant of the strain rate tensor (Eq. 9) that does not include vertical shear terms:

$$\dot\varepsilon_d^2=\left(\frac{\partial u_b}{\partial x}\right)^2+\left(\frac{\partial v_b}{\partial y}\right)^2+\frac{\partial u_b}{\partial x}\frac{\partial v_b}{\partial y}+\frac{1}{4}\left(\frac{\partial u_b}{\partial y}+\frac{\partial v_b}{\partial x}\right)^2.$$

(18)

In Yelmo, $\dot\varepsilon_d$ is only used for calculating $\eta_d$, while the 3D effective strain rate is calculated from the full strain rate tensor in the material component (see Material section above). Calculating the full tensor during the iterative SSA solution procedure would be much more computationally expensive, while the 2D effective strain rate is already sufficient for the vertically integrated

SSA equations (Pollard and DeConto, 2012).

The stress boundary condition imposed at the floating ice front, following Winkelmann et al. (2011) and Greve and Blatter (2009), is

$$\eta_d\left(4\frac{\partial u}{\partial x}+2\frac{\partial v}{\partial y}\right)n_x+\eta_d\left(\frac{\partial u}{\partial y}+\frac{\partial v}{\partial x}\right)n_y=\left(\frac{1}{2}\rho gH^2-\frac{1}{2}\rho_{\mathrm{sw}}gH_{\mathrm{o}}^2\right)n_x$$
$$\eta_d\left(4\frac{\partial v}{\partial y}+2\frac{\partial u}{\partial x}\right)n_y+\eta_d\left(\frac{\partial v}{\partial x}+\frac{\partial u}{\partial y}\right)n_x=\left(\frac{1}{2}\rho gH^2-\frac{1}{2}\rho_{\mathrm{sw}}gH_{\mathrm{o}}^2\right)n_y.$$

(19)

The depth of the seawater up to the flotation depth, $H_{\mathrm{o}}$, is defined as: $H_{\mathrm{o}}=\min\left(\max\left(z_{\mathrm{sl}}-z_b,0\right),\frac{\rho}{\rho_{\mathrm{sw}}}H\right)$. This is the depth

of the ocean directly adjacent to the ice sheet, which acts to reduce the outward pressure at the floating ice margin. In constrast to Winkelmann et al. (2011), this boundary condition is not currently used in Yelmo for grounded ice, where Eq. 16 applies.

The SSA equations are nonlinear, elliptical, partial differential equations with non-local solutions. Yelmo uses Lis for the numerical solution using the biconjugate gradient method. The subroutine to discretize the equations and to call Lis was ported from the latest SICOPOLIS version 5-dev (Greve, 2019; Rückamp et al., 2019) and subsequently modified for model design

choices in Yelmo. We use a Picard iteration method to account for the nonlinear dependence of the effective viscosity ($\eta_d$), and potentially the basal friction coefficient ($\beta$), on velocity. Convergence of the SSA solution is tested using the $L^2$ relative error

norm (Gagliardini et al., 2013):

$$\delta_{\mathrm{u,v}} = \frac{2\sqrt{\sum (u_1 - u_0)^2 + \sum (v_1 - v_0)^2}}{\sqrt{\sum (u_1 + u_0)^2 + \sum (v_1 + v_0)^2}}, \tag{20}$$

where $(u_1, v_1)$ and $(u_0, v_0)$ are the velocity solutions for the current and previous iterations, respectively, and the sum is made over all grid points with non-zero velocity being considered by the SSA solver. By default, we consider a convergence limit of $\delta_{\mathrm{u,v}} = 10^{-2}$, which is typically achieved within 1-10 iterations, depending on the context. This limit can be specified by the user.

The result of solving the above equations is the hybrid, 3D horizontal velocity field $(u, v)$. The vertical velocity $w$ can then be diagnosed by applying a kinematic boundary condition at the base, and integrating the continuity equation for incompressible flow (Greve and Blatter, 2009), from $z_b$ to $z$,

$$w(z) = \dot{b} - \left( u_b \frac{\partial z_b}{\partial x} + v_b \frac{\partial z_b}{\partial y} \right) - \int_{z_b}^{z} \left( \frac{\partial u}{\partial x} + \frac{\partial v}{\partial y} \right) dz'. \tag{21}$$

The vertical velocity is naturally defined on the ac-nodes in the vertical plane, analogous to the horizontal velocity in the horizontal plane. The above dynamics update results in a 3D hybrid velocity field $(u, v, w)$.

Basal frictional stress, as it appears in the SSA elliptical equations, is defined as

$$\tau_b = -\beta \mathbf{u}_b \tag{22}$$

where $\beta$ represents the basal friction coefficient, with units of $[\mathrm{Pa\,yr\,m^{-1}}]$, which can be defined in several ways. $\beta$ is prescribed to be zero for floating ice, and otherwise can be set to a constant or a spatially varying field and, depending on the formulation used, it can depend on velocity itself. For this reason, we also define $c_b$ as the bed friction coefficient, which we consider to only provide information about conditions at the physical bed (e.g., the nature of basal sediments, basal hydrology, effective pressure, etc.), independent of velocity. In the model, therefore, $\beta$ is defined as:

$$\beta = c_b f(\mathbf{u}_b). \tag{23}$$

Thus in all formulations implemented in Yelmo, the term $f(\mathbf{u}_b)$ has units of $[\mathrm{yr\,m^{-1}}]$ and the coefficient $c_b$ has units of $[\mathrm{Pa}]$, which helps to facilitate its physical interpretation.

Most commonly, $\beta$ is defined using a linear (e.g. Quiquet et al., 2018), power-law (e.g. Pattyn, 2017), pseudo-plastic power-law (e.g. Aschwanden et al., 2013) or regularized-Coulomb (Joughin et al., 2019) formulation. The linear and power-law formulations are contained within the pseudo-plastic power-law formulation, so only the latter and the regularized-Coulomb formulation are needed to represent all four cases.

The pseudo-plastic power-law formulation (Schoof, 2010; Aschwanden et al., 2013) is

$$\tau_b = -c_b \left( \frac{|\mathbf{u}_b|}{u_0} \right)^q \frac{\mathbf{u}_b}{|\mathbf{u}_b|} \tag{24}$$

and thus $\beta = c_b u_0^{-q} |\mathbf{u}_b|^{q-1}$, with the pseudo-plastic exponent $q \in (0,1)$ and threshold speed $u_0$. This expression results in purely plastic friction for $q = 0$, linear friction for $q = 1$ and power-law friction for $0 < q < 1$. With $q = 1$ and $u_0 = 1$, for example, $\beta = c_b$ and friction scales linearly with velocity. To obtain the power-law formulation used in the original MISMIP experiments (Pattyn et al., 2012), the following parameter values can be prescribed: $q = 1/3$, $u_0 = 1\,\mathrm{m\,yr^{-1}}$ and

$c_b = 3.165176 \times 10^4\,\mathrm{Pa}$.

Alternatively, the regularized Coulomb law (Schoof, 2005; Brondex et al., 2019; Joughin et al., 2019) is defined as

$$\tau_b = -c_b \left( \frac{|\mathbf{u}_b|}{|\mathbf{u}_b| + u_0} \right)^q \frac{\mathbf{u}_b}{|\mathbf{u}_b|} \tag{25}$$

and thus $\beta = c_b(|\mathbf{u}_b| + u_0)^{-q} |\mathbf{u}_b|^{q-1}$. Again $q$ is the non-linear exponent and $u_0$ is an empirical threshold speed that dictates the transition from Coulomb friction when cavitation effects dominate at the base (typically for a hard bed) to Coulomb-plastic

friction, when friction saturates (typically for weak till). When $u_0 = 0$ or $q = 0$, purely plastic friction is recovered.

The merits and physical basis of the different possible friction formulations and non-linear exponents are still under active debate (Aschwanden et al., 2013; Stearns and van der Veen, 2018; Brondex et al., 2019; Joughin et al., 2019), and all of the above formulations are used in ice-sheet modeling today. However, given the large uncertainty in boundary conditions provided to an ice-sheet model, which include bedrock topography, sediment composition and distribution, basal hydrology

and its temporal evolution, etc., it is clear that the use of any formulation will rely on empirical tuning. Also, as noted above, different choices for the friction exponents or threshold values can reduce a given formulation to another. Although modeling studies have shown that all four cases above can produce realistic velocity fields of the present-day ice sheets (e.g. Goelzer et al., 2018; Joughin et al., 2019), it remains to be seen how the choice of friction formulation may impact transient changes in the ice sheet.

For these reasons, we have chosen to implement the friction formulations in the most general way possible in the code, with essentially two free parameters: $q$ as a non-linear exponent and $u_0$ as a threshold speed. Meanwhile, $c_b$ is a 2D field that can be set to a constant value, or a spatially and/or temporally varying field based on e.g., whether the ice is frozen to the bed or temperate, on till strength (Bueler and van Pelt, 2015), effective pressure, or other user-defined criteria. As mentioned above, a Picard iteration method is used to solve for basal friction, $\eta_d$ and the SSA velocity solution until convergence of the velocity

solution is reached.

$\beta$ and $c_b$ are initially defined on aa-nodes. $c_b$ is naturally defined on the grid center, while when $\beta = f(\mathbf{u}_b)$, the velocity components that are defined on ac-nodes must be staggered to the grid center. Once $\beta$ has been calculated using one of the above formulations, it must be staggered to the ac-nodes for use in the elliptical solver. For purely floating points (i.e., $f_g = 0$ at both bounding aa-nodes) $\beta_{\mathrm{ac}} = 0$, and for purely grounded points, $\beta_{\mathrm{ac}}$ is the average of the two neighbors. At the grounding

line, Yelmo allows several options to handle staggering. These include simple averaging, taking the upstream value of $\beta$, taking the downstream value of $\beta$ or taking the weighted average based on the grounded fraction of the ac-node.

### 3.4 Thermodynamics

Thermodynamics in Yelmo is treated in the classical way by solving the following energy conservation equation:

$$\frac{\partial T}{\partial t} = \frac{k}{\rho c} \frac{\partial^2 T}{\partial z^2} - u \frac{\partial T}{\partial x} - v \frac{\partial T}{\partial y} - w \frac{\partial T}{\partial z} + \frac{\Phi}{\rho c} \tag{26}$$

where $k$ and $c$ are the ice thermal conductivity and specific heat capacity, respectively. The evolution of the ice temperature $T$ is driven by vertical diffusion, horizontal and vertical advection, and internal strain heating due to ice shearing, $\Phi$, where

$$\Phi = 4\eta\dot{\varepsilon}^2. \tag{27}$$

Horizontal diffusion is assumed to be negligible (Greve and Blatter, 2009). At the air-ice interface (i.e., the ice surface), the ice temperature is prescribed via the input boundary temperature field $T_s$, limited to a maximum value of $T_0 = 273.15K$. At the base of floating ice, the ice temperature is prescribed to the expected freezing temperature of seawater as a function of depth (Jenkins, 1991), except near the grounding line, where the temperature is prescribed to the pressure melting point of ice. At the base of grounded ice, when the ice temperature is below the pressure melting point, the vertical gradient of temperature is prescribed as $\partial T/\partial z = -Q_{\mathrm{geo}}/k$, where the geothermal heat flux ($Q_{\mathrm{geo}}$) is provided as a boundary field to Yelmo. If the temperature at the ice base reaches the pressure melting point, then the temperature is set to the pressure melting point, and the basal mass balance is diagnosed as (Cuffey and Paterson, 2010):

$$\dot{b}_{\mathrm{g}} = -\frac{1}{\rho L}\left(Q_b + k\frac{\partial T}{\partial z}\bigg|_b + Q_{\mathrm{geo}}\right) \tag{28}$$

where $\dot{b}_{\mathrm{g}}$ is the basal mass balance of grounded ice (negative for melting), $L$ is the latent heat of fusion for ice, $Q_b$ is the basal heat production to due sliding and $\frac{\partial T}{\partial z}\big|_b$ is the ice temperature gradient at the base. Yelmo calculates $\dot{b}_{\mathrm{g}}$, which is a model output, in contrast to $\dot{b}_{\mathrm{f}}$ (basal mass balance of floating ice), which is prescribed in Yelmo as a boundary condition. Once the ice base is temperate (i.e., at the pressure melting point), it will remain so as long as $W_{\mathrm{til}} - \left(\frac{\rho_w}{\rho}\dot{b}_{\mathrm{g}}\right)\Delta t > 0$, where $\dot{b}_{\mathrm{g}}$ is used from the previous timestep and $W_{\mathrm{til}}$ is the water layer thickness in the till beneath the ice sheet. In other words, if it is expected that an energy deficit will result in freeze-on of the total available liquid water at the ice base, then the point is treated as a non-temperate ice point.

Yelmo simulates the evolution of the basal water layer thickness in the till following Bueler and van Pelt (2015):

$$\frac{\partial W_{\mathrm{til}}}{\partial t} = -\frac{\rho}{\rho_w}\dot{b}_{\mathrm{g}} - C_d, \tag{29}$$

where $C_d$ is the prescribed till drainage rate, usually set to $C_d = 0.001\,\mathrm{m\,yr}^{-1}$. $W_{\mathrm{til}}$ is limited to the range $0 \leq W_{\mathrm{til}} \leq W_{\mathrm{til,max}}$ where maximum is usually set to $W_{\mathrm{til,max}} = 2\,\mathrm{m}$. This approach allows for $W_{\mathrm{til}}$ to maintain consistency with the thermodynamic state of the ice sheet at all times. It does not include horizontal transport, as this could potentially be treated by an external basal hydrology model. It is also possible to disable calculation of $W_{\mathrm{til}}$ inside of Yelmo, and instead consider it as a boundary variable. However, given the adaptive timestepping approach used by Yelmo, we have found that updating $W_{\mathrm{til}}$

internally at each timestep helps to avoid artificial oscillations that may develop otherwise when the thermodynamics and basal friction are coupled.

Eq. 26 is solved with an implicit method in the vertical direction, while the horizontal advection is solved separately applying an explicit, second-order upwind forward Euler method. This separation allows the energy conservation in the vertical to be solved as a 1D column model. The discretization of vertical diffusion follows the form presented by Hoffman et al. (2018), while the discretization of vertical advection follows a second-order central difference scheme. A given column of grid points consists of temperatures defined on the grid-centers (aa-nodes) and boundary values defined directly at the surface and base of the ice sheet.

## 4 Timestepping

Yelmo makes use of a predictor-corrector (PC) method combined with adaptive timestepping to balance speed and stability, following the method developed by Cheng et al. (2017). This approach requires calculating the ice thickness twice per timestep, while all other variables can be calculated only once per timestep. Applying a PC method significantly improves the accuracy of the solution compared to a simple forward Euler timestepping method. Furthermore, it facilitates the calculation of a stability metric at each timestep that can be used to evaluate model performance and forms the basis of a robust adaptive timestepping approach (Cheng et al., 2017). A given timestep therefore consists of three parts:

1. **Predictor step**: The topography component (namely the ice thickness) is predicted using the dynamics, material and thermodynamics solutions from previous timesteps.

2. **Update step**: Using the predicted topography solution, the dynamics, material and thermodynamics components are then also updated.

3. **Corrector step**: Using the updated dynamics, material and thermodynamics component solutions, the topography component is finally calculated again, starting from the ice thickness solution of the previous timestep.

In Yelmo, the predictor step is calculated via the second-order Adams-Bashforth (AB) method (Cheng et al., 2017),

$$H_{n+1}^{\star} = H_n + \Delta t_n \left[ \beta_1 f \left( H_n, \bar{\mathbf{u}}_n \right) + \beta_2 f \left( H_{n-1}, \bar{\mathbf{u}}_{n-1} \right) \right], \tag{30}$$

where $H^{\star}$ is the predicted ice thickness, $\Delta t$ is the timestep, and $\beta_1 = 1 + \frac{\zeta_t}{2}$, $\beta_2 = -\frac{\zeta_t}{2}$ and $\zeta_t = \frac{\Delta t_n}{\Delta t_{n-1}}$. The labels $n$, $n-1$ and $n+1$ indicate the current, previous and next timestep, respectively. Here, $f \left( H, \bar{\mathbf{u}} \right)$ is shorthand for $\frac{\partial H}{\partial t}$ as a function of the ice thickness and depth-averaged horizontal velocity field, noting that $\bar{\mathbf{u}}$ is also a function of the ice thickness, material and potentially thermodynamic state of the ice sheet. For this algorithm, $\beta_1$, $\beta_2$ and $\zeta_t$ are timestep dependent, but the subscript $n$ has been dropped for clarity. Once $H_{n+1}^{\star}$ has been calculated, the other components are updated, and finally the corrector step is then calculated via the Semi-implicit Adams-Moulton (SAM) method (Cheng et al., 2017),

$$H_{n+1} = H_n + \frac{\Delta t_n}{2} \left[ f \left( H_{n+1}^{\star}, \bar{\mathbf{u}}_{n+1} \right) + f \left( H_n, \bar{\mathbf{u}}_n \right) \right], \tag{31}$$

where $H_{n+1}$ is the corrected ice thickness for the next timestep.

For the AB-SAM timestepping method, Cheng et al. (2017) have derived the following expression for the leading term of the local truncation error:

$$\tau_{n+1} = \frac{\zeta_t \left( H_{n+1} - H_{n+1}^{\star} \right)}{(3\zeta_t + 3)\, \Delta t_n} \tag{32}$$

The local truncation error is valuable for diagnosing the performance of the model, and can be used as an indicator of numerical stability. For a small enough timestep, $H_{n+1}^{\star}$ and $H_{n+1}$ should be indistinguishable and $\tau^{n+1} \sim 0$. However, as the timestep increases, the local truncation error will also increase.

An adaptive timestepping approach based on a proportional-integral (PI) controller method is therefore used to maximize the timestep while maintaining the truncation error below a specified threshold (Cheng et al., 2017; Söderlind and Wang, 2006).
Defining the maximum truncation error over all grounded grid points as $\eta = \max|\tau|$, the next timestep is calculated using the so-called PI4.2 controller Söderlind (2002):

$$\Delta t_{n+1} = \left( \frac{\epsilon}{\eta^{n+1}} \right)^{(k_I + k_p)} \left( \frac{\epsilon}{\eta^n} \right)^{-k_p} \Delta t_n, \tag{33}$$

where $\epsilon$ is the target tolerance and $k_I = 2/10$ and $k_p = 1/10$ are reasonable control parameters for the second-order AB-SAM timestepping method used here (Söderlind and Wang, 2006). This algorithm ensures that the time step increases when $\eta < \epsilon$
and decreases when $\eta > \epsilon$. The use of both $\eta^{n+1}$ and $\eta^n$ helps to avoid rapid fluctuations in the timestep, which improves model stability and results in a predictable timestep size as a function of the target tolerance.

For practical purposes, the timestep is further treated as follows. The timestep must be larger than a user-prescribed minimum value, but smaller than the Courant–Friedrichs–Lewy (CFL) 2D advective limit:

$$\Delta t_{\mathrm{cfl}} = C_{\mathrm{cfl}} \max \left| \frac{\bar{u}}{\Delta x} + \frac{\bar{v}}{\Delta y} \right|^{-1} \tag{34}$$

where the maximum is taken over all grid points and $C_{\mathrm{cfl}} = 1.0$. Furthermore, the adaptive timestep is adjusted to ensure that the model stays synchronized with the frequency that Yelmo is being called externally. We found that the latter requirement often results in highly uneven timestepping; e.g., if Yelmo is called with a timestep of $\Delta t_{\mathrm{tot}} = 2.0\,\mathrm{yr}$ and the first adaptive timestep is determined to be $\Delta t_1 = 1.9\,\mathrm{yr}$, then the second timestep would likely be $\Delta t_2 = 0.1\,\mathrm{yr}$. To avoid this possibility and increase stability, the condition is imposed that if any given adaptive timestep is predicted in the range of $0.5\Delta t_{\mathrm{tot}} < \Delta t < \Delta t_{\mathrm{tot}}$, then
$\Delta t = 0.5\Delta t_{\mathrm{tot}}$. In this example, this condition would ensure that $\Delta t_1 = \Delta t_2 = 1.0\,\mathrm{yr}$, unless $\Delta t_2$ needed to be smaller for stability. Finally, if the maximum local truncation error $\eta$ is larger than a specified threshold for any given integration, then the integration is discarded and repeated with a progressively smaller timestep until $\eta$ diminishes and stability is restored, or the timestep reaches the minimum allowed value.

## 5 Model interface

The Yelmo model interface is designed to be clear and simple, but also flexible. In its essence, there are three main model functions: `yelmo_init` to initialize the model variables, `yelmo_update` to perform the ice-sheet model calculations for a given timestep and `yelmo_end` to terminate the Yelmo object (free it from memory).

The first subroutine, `yelmo_init`, is used to load parameters, initialize variables in memory (i.e., allocate arrays) and, optionally, to initialize the topographic state variables (ice thickness, masks, etc.). No other variables are initialized here in the sense of being populated with data values, which is left to the user. An additional, optional helper function can be used, `yelmo_init_state`, which populates the remaining model variables in the material, thermodynamics and dynamics components. This initialization step is separated from that of topography because in practice, sometimes boundary variables

(e.g., surface temperature) need the surface elevation as input in order to be determined. In contrast, the remaining variables, namely dynamics and thermodynamics, often rely on boundary variables to be initialized. Thus, a typical initialization sequence for a stand-alone ice-sheet model simulation could first call `yelmo_init`, then load or calculate boundary variables and then call `yelmo_init_state` to finalize the initialization of all Yelmo variables. After this sequence, the Yelmo state should be consistent with running the model for one timestep with the prescribed boundary conditions and a fixed topography. If the

model will be initialized from a restart file, then these data are loaded in each case based on parameter choices.

The next subroutine, `yelmo_update`, is used to advance the model state to a new timestep. Any modifications to boundary variables are left to the user externally, and Yelmo expects that the boundary conditions are valid for this timestep. The sub-routine does not take any arguments to modify the model behavior – rather, all model configuration choices are specified in the parameters of the Yelmo object itself. These are initially loaded from a parameter file in the call to `yelmo_init`, however,

it is possible to modify any parameter values during simulations, allowing for changing the model configuration transiently depending on the experimental setup. An additional optional subroutine, `yelmo_update_equil`, is available to facilitate equilibration. This routine effectively calls `yelmo_update` for a specified time window with unchanging boundary conditions, and allowing for the temporary modification of some key model parameters (such as the maximum allowed adaptive time step and the maximum allowed SSA velocity).

The last subroutine, `yelmo_end`, simply removes the Yelmo object from memory (i.e., all domain variables are deallocated). After calling `yelmo_end`, it is possible to reinitialize the Yelmo object via `yelmo_init`, for example, in order to test a different grid resolution or other configuration.

There are several input/output routines defined for Yelmo. `yelmo_write_init` can be used to initialize a NetCDF model output file with the axes of model dimensions defined from the Yelmo object and writing of static fields like domain masks. The

writing of model output for inidividual timesteps is left to the user to maintain flexibility, as most programs require specific fields to be written (examples can be found in the test programs included with the code - see further below). In addition, `yelmo_restart_write` will create a NetCDF file and write all Yelmo fields as a snapshot, which can be used to restart the model (loading of a restart file can be activated with parameter choices).

As mentioned above, given the object-oriented approach, it is possible to run multiple Yelmo domains in one program. Each domain must be initialized separately via `yelmo_init`, and the variable fields populated with initial values, then separate calls to `yelmo_update` are needed during timestepping, and finally each object should be terminated at the end of the program via `yelmo_end`. With this structure, minimum modification of another model, like a global climate model is needed, to incorporate online ice-sheet evolution, or to simulate an ensemble of ice sheets in one program. Furthermore, all fields are directly accessible within the main program to facilitate coupling. For example, the 2D array of surface elevation of the topography component of the Yelmo Antarctica domain could be referenced as `yelmo_ant%tpo%now%z_srf`. While it is clear that the nesting of several containers (derived types) results in a rather long variable reference, it is unambiguous and straightforward to use.

## 6   Model validation and benchmarks

Yelmo has been tested against several ice-sheet model validation tests and benchmarks in wide use today. These include the Halfar dome experiment (Halfar, 1983; Bueler et al., 2005), the EISMINT1 (Huybrechts et al., 1996) and EISMINT2 (Payne et al., 2000) model intercomparison experiments that test uncoupled and coupled dynamics-thermodynamics, respectively, and MISMIP (Pattyn et al., 2012) for ice-shelf dynamics, among others. By design, many of these experiments allow isolation of specific model features for testing. When the model passes more complex benchmark tests, the simpler experiments are somewhat redundant (if the model passes a coupled thermodynamics-dynamics benchmark, the model should necessarily also be able to pass a dynamics-only benchmark). However, it should be noted that in the process of model development, all tests prove to be extremely useful. The results of all tests will not be reported here, but several are highlighted below to demonstrate that Yelmo performs well.

The Halfar dome experiment, a specific case of the more general Test B of Bueler et al. (2005), tests the ice-sheet model dynamics using the SIA solver alone. This test consists of simulating a radially-symmetric ice-sheet with zero mass balance and resting on a flat bed, deforming under gravitational stress. The analytical solution is known at every time, allowing a direct comparison of the simulation to the desired result. The simulation parameters consist of the margin radius and dome elevation, in this case set to the values suggsted by Halfar (1983): $R_0 = 21.2132$ km and $H_0 = 707.1$ m – see Bueler et al. (2005) for further details. Figure 3 shows the root mean square error (RMSE) of the simulation with the analytical result after 200 years for a range of model resolutions. Yelmo demonstrates first-order ($p = 1.01$) numerical convergence with resolution towards the analytical result.

The EISMINT1 moving margin experiment also tests the ice-sheet model dynamics using the SIA solver alone, with an imposed constant rate factor and diagnosed thermodynamics (i.e., thermodynamics do not impact the ice-sheet configuration). Radial steady-state surface mass balance and background surface temperature fields are imposed as boundary conditions. Starting from ice-free conditions, the ice sheet simulated by Yelmo grows to dynamic and thermodynamic equilibrium within 25 kyr and 100 kyr, respectively. The steady-state summit elevation of Yelmo is 3006.6 m compared to the reported range of 2997.5±7.4 m for so-called "Type-I" discretization models like Yelmo (where diffusivity is staggered to the ab-nodes). The

basal temperature relative to the pressure melting point (i.e., homologous temperature) at the summit simulated by Yelmo is -13.37 °C, which lies within the EISMINT1 range of -13.40±0.56 °C. These and other relevant statistics are given in Table 1.

We also use the EISMINT1 moving margin experiment to demonstrate the capability of the adaptive timestepping approach in Yelmo. By setting the tolerance parameter $\epsilon$, Yelmo automatically adjusts the timestep to maintain the truncation error in ice thickness $\eta$ around this value. Figure 4a shows the time series of the adaptive timestep used by Yelmo for a 25 kyr simulation for different resolutions. The timestep exhibits oscillations around a mean value, which is typical for such a PID approach (Cheng et al., 2017). When the timestep grows larger, the truncation error increases. This leads to a reduction in the timestep and the error decreases. Figure 4b shows the mean timestep used by Yelmo over the last 10 kyr of the simulation versus model resolution. Given a tolerance of $\epsilon = 10^{-2}$, Yelmo's mean timestep is $\Delta t = 6.96$ yr, $\Delta t = 1.59$ yr, $\Delta t = 0.24$ yr and $\Delta t = 0.06$ yr for resolutions of 50 km, 25 km, 10 km and 5 km, respectively. As expected, the timestep must be reduced for higher resolutions. These results are in line with those of Cheng et al. (2017) for the same experiment ($\Delta t = 12.4$ yr for 60 km resolution). It should be noted that the truncation error increases non-linearly as a function of the timestep, so setting a higher tolerance does not translate directly into a larger timestep.

Next we validate the thermodynamics component first by performing the benchmark experiments Test A and Test B of Kleiner et al. (2015). In contrast to an enthalpy solver, Yelmo uses a temperature solver that assumes all water produced in the ice column drains directly to the bed and so temperate ice in the vertical column has no water content. In cases where water content of up to 3 % could be present in the basal layers of the column, Yelmo's solver would be inaccurate. Nonetheless, we expect that the temperature solver should be sufficiently accurate to simulate ice sheets on long timescales and large spatial domains. Figure 5 shows the performance of Yelmo's temperature solver for Test A of Kleiner et al. (2015), which simulates a column of ice in a parallel-sided slab with no horizontal advection and no internal strain heating that undergoes warming and subsequent cooling at the surface. In this case, no water content should develop in the vertical column, so a temperature and enthalpy solver should give identical, energy-conserving results. Yelmo's basal melt rate is essentially identical to the analytical solution for this problem and its transient behaviour is robust.

In contrast, Fig. 6 shows the results of Yelmo's temperature solver for Test B, which simulates a parallel-sided slab on a sloping bed with a prescribed horizontal velocity and strain heating profile in steady state. In this case, water is generated in the basal layers of the ice column (see Fig. 6c), however Yelmo cannot reproduce this solution. Nonetheless, because temperature is limited to the pressure-melting point, the simulated ice temperature profile is in full agreement with the analytical profile. This is true both for a very high resolution case ($\Delta z = 0.5$ m) and a lower resolution case ($\Delta z = 10$ m), which allows us to conclude that the performance of the temperature solver is robust.

The EISMINT2 benchmark experiments A and F are useful for testing the thermodynamically coupled ice-sheet model with SIA dynamics like in EISMINT1. The experiments are identical to the EISMINT1 moving margin experiment, except the resolution is doubled (25 km) and the surface temperature is prescribed to be independent of ice thickness. Experiment A prescribes a summit temperature of 238.15 K, while experiment F is 15.00 K colder, which promotes an increase in the region of ice frozen to the bedrock. The statistics for these experiments are listed in Table 1 as well. Figure 7 shows the basal homologous temperature distribution for experiments A and F. Yelmo produces temperature patterns in both experiments,

which are consistent with both the benchmark results (Payne et al., 2000) and other more recent models (e.g., Bueler et al., 2007; Hoffman et al., 2018). Axial symmetry, assessed by comparing the basal temperature field with a mirror of itself along the x- or y-axis, is maintained to a precision of $10^{-2}$ K. This symmetry is not critical to realistic applications, but a lack of at least axial symmetry in this test is often indicative of numerical artifacts. In experiment F, Yelmo produces the so-called "cold spokes", which have been shown to be related to internal strain heating in regions of steep gradients in ice thickness, and largely numerical in nature (Bueler et al., 2007).

We also test the capability of the SSA solver and grounding-line treatment by running the MISMIP protocol experiments (Pattyn et al., 2012). Particularly, MISMIP EXP 1 (advance) and EXP 2 (retreat) are useful for testing the reversibility of grounding line advance, given the bedrock is defined as a linear downward sloping bed. The rate factor is prescribed according to steps that first decrease, allowing grounding line advance, then increase back to the original value. According to theory (Weertman, 1974; Schoof, 2007), only one steady-state grounding line position should exist for each step - i.e., the ice sheet should advance and retreat symmetrically without showing hysteresis. It is now well known, however, that ice-sheet models at coarse resolutions (1 km and greater) are unable to capture proper grounding-line migration, even when subgrid parameterizations to mimic higher resolution are applied (Seroussi et al., 2014; Gladstone et al., 2017).

In the MISMIP experiment performed here, the linear, downward sloping bedrock is defined in the x-direction as $z_b = 720.0 - 778.5\,(x/750.0)$ with $x$ in km and $z_b$ in m. The bedrock elevation does not change in the y-direction, which extends to $\pm 50$ km to allow the simulation of a symmetric ice stream flowing in the positive x-direction. The power-law formulation of Eq. 24 is used with the parameter values $q = 1/3$, $u_0 = 1\,\mathrm{m\,yr^{-1}}$ and $c_b = 3.165176 \times 10^4$ Pa. The rate factor is initially prescribed to $A = 1 \times 10^{-16}\,\mathrm{Pa^{-3}\,yr^{-1}}$ and the simulation is run for 25 kyr to equilibrate. Next, the rate factor is stepped evenly in log-space every 10 kyr until reaching $A = 1 \times 10^{-19}\,\mathrm{Pa^{-3}\,yr^{-1}}$, and then the rate factor is increased in the same way until returning to the original value.

Figure 8 shows results for this MISMIP experiment with Yelmo at different grid resolutions, ranging from 20 km down to 2.5 km, and with different treatments of basal friction near the grounding line. When the default model setup is used, with no special treatment at the grounding line, the grounding line advance is consistent for all resolutions. However, none of the lowest resolution simulations show grounding line retreat as the rate factor increases again. At a resolution of 5 km, some minor grounding line retreat can be seen, and for 2.5 km, the model is more successful at retreating though it remains 400 km from the target. In contrast, when the basal friction $\beta$ is scaled at the grounding line by the grounded fraction of the ac-node ($f_{\mathrm{g,ac}}$), the hysteresis is greatly reduced. The 5 km simulation retreats to within 200 km of the original position and the 2.5 km simulation retreats to within 100 km of the original position, thus showing convergence to the correct solution with resolution. With this setup, even the 10 km and 20 km simulations retreat significantly. In a third case, the basal friction is also linearly scaled to zero as the ice sheet approaches flotation (Leguy et al., 2014; Gladstone et al., 2017). In this case, the hysteresis and differences between different resolution simulations are further reduced, however, the system also tends to advance much less given all other conditions are the same.

Yelmo's Eulerian conservative tracer model is validated with a simulation of ice age in an idealized cofiguration against the analytical solution presented by Rybak and Huybrechts (2003). In this case, summit-like conditions are imposed, in that

horizontal advection is neglected, and the vertical velocity is assumed to decrease linearly with depth. Figure 9 shows the solution with Yelmo as compared to the analytical result. For a nominal vertical resolution of $n_z = 30$ points and single or double precision, the age tracer gives errors in the range of 0.2-0.5% over most of the column of the ice sheet, with higher errors at the base. Increasing the vertical resolution to $n_z = 50$ points decreases the error by an order of magnitude, while using 5  $n_z = 30$ with higher resolution at the base of the ice sheet allows a similar reduction in error with significant computational savings. For Eemian-age ice in such simplified conditions, the latter case gives an uncertainty of less than 1 kyr. It is expected that the error would increase for more realistic 3D domains, however the Eulerian age solver can be used for a first-order estimate of the age-depth profile in the ice sheet (Rybak and Huybrechts, 2003).

## 7  Antarctica

10  As further validation of the model's performance, we ran steady-state simulations of the present-day and glacial Antarctic ice sheet. These simulations, run at 32 km resolution, have been deliberately simplified to include the minimum complexity necessary to simulate the ice sheet without additional external components. There was no active isostasy model and geothermal heat flux was set to 50 mW m$^{-2}$ everywhere. Basal friction followed a linear law, where $\beta = (c_f \lambda_b / u_0)(\rho g H)$ with $u_0 = 100$ m yr$^{-1}$ used as a scaling constant. We prescribed $c_f = 0.15$ (unitless) for most of the domain, except for ad-hoc adjustments 15  in specific regions to improve the match with observations. This was additionally scaled by an exponential function of bedrock elevation: $\lambda_b = \min[1.0, \exp((z_b - z_1)/(z_1 - z_0))]$, analogous to the approach of (Martin et al., 2011). We set $z_1 = 250$ m everywhere and $z_0 = -2000$ m for WAIS regions feeding the Ronne ice shelf and $z_1 = -200$ m elsewhere. Friction was scaled by the grounded fraction at the grounding line, but no additional scaling is applied. The enhancement factor parameters set for these simulations were $E_{\text{shr}} = 2.5$, $E_{\text{strm}} = 0.7$ and $E_{\text{shlf}} = 0.5$. The bedrock topography and initial ice thickness were 20  prescribed from the RTOPO2.1 dataset (Schaffer et al., 2016), after which the model ran for 50 kyr, reaching a steady-state modeled ice distribution.

For the simulation of the present-day state, surface mass balance (SMB) and surface temperature boundary fields were prescribed from a RACMO2.3 simulation driven by ERA-INTERIM data and averaged over 1981-2010 (van Wessem et al., 2018). The ice-shelf basal mass balance was set to a spatially constant value of -0.2 m a$^{-1}$ where floating ice exists today and 25  to -2.0 m a$^{-1}$ elsewhere. Figures 10 and 11 show a comparison of the simulations with the observed topography (RTOPO2.1) and the present-day observed velocity (Rignot et al., 2011). With this relatively simple model setup, it is possible to obtain reasonable agreement with observations. The root mean square errors (RMSEs) in ice thickness, velocity and log(velocity) are 320 m, 270 m yr$^{-1}$ and 1.9 log[m yr$^{-1}$], respectively, which fall in the range of other models in the initMIP-Antarctica intercomparison project (Seroussi et al., 2019). The simulated ice sheet is thinner than the observed ice sheet over large parts 30  of East Antarctica, with a broad positive bias near the South Pole (Fig. 11). The margins of the ice sheet are the most difficult to match and, in particular, the grounding-line positions of the large ice shelves, leading to larger biases in these regions. This pattern is quite consistent with other studies (e.g., Martin et al., 2011; Quiquet et al., 2018; Albrecht et al., 2020). Overall, the

dome configuration, slow deformational speeds and even most ice streams as they penetrate inland are well represented by the model (Figs. 10 and 11).

We use the same setup with modified boundary conditions to simulate a configuration resembling that of a deep glacial period like the Last Glacial Maximum. The surface temperature was set to $10°$ C colder and the present-day SMB was maintained, except for points with a low or negative SMB were prescribed with a minimum value of $0.1\,\mathrm{m\,a^{-1}}$. The ice shelf basal mass balance was set to a spatially constant value of $0\,\mathrm{m\,a^{-1}}$ and sea level was lowered by $120\,\mathrm{m}$. In this case, the grounded ice sheet advances until the continental shelf break and thickens inland (Fig. 10). A similar structure of ice streams can be seen, due to the topographic dependence of $\beta$, but their speed is greatly reduced compared to those of the present-day simulation. We do not expect this configuration to be realistic, given that isostasy plays no role and a present-day-like SMB has been imposed. However, this test demonstrates that Yelmo is capable of resolving continental-scale changes in the ice sheet configuration in a plausible way.

## 8  Conclusions and future work

We have described the features and physics of the hybrid ice-sheet–shelf model Yelmo. Yelmo includes the physics to simulate continental-scale ice sheets and floating ice shelves using "shallow" approximations of the ice dynamics. The fully coupled thermomechanical ice-sheet model has been validated against several benchmark tests, and has been shown to simulate the dynamic configuration of the Antarctic ice sheet well.

Yelmo is expected to be useful for long-time scale simulations and/or ensembles. It is particularly suited for easy coupling with other models. For example, the simulation of multiple ice-sheet domains with independent parameter configurations coupled to a global climate model can be achieved in a simple and straightforward way. Also, given that the subroutines representing the physics of the model have been isolated from the "model accounting", it is possible to test individual model components in different contexts easily. This should facilitate future model development and comparison of different methods.

The model framework has been designed to facilitate the incorporation of new and different physics. Thus, this initial release of Yelmo lays the foundation for several future developments. These may include more advanced calving and basal friction schemes, as well as improved treatment of the grounding line. We also plan to transition to an enthalpy-based thermodynamics solver, however this will require an adaptive vertical axis to be able to map the height of transition between temperate and cold ice accurately. We also plan to implement a variationally-derived "depth-integrated-viscosity approximation" solver (following e.g., Goldberg, 2011; Pollard and DeConto, 2012; Lipscomb et al., 2019) in the near future.

*Code availability.* Yelmo is maintained as a git repository hosted at https://github.com/palma-ice/yelmo under the licence GPL-3.0. Model documentation can be found at https://palma-ice.github.io/yelmo-docs/. The exact version of the model, along with the necessary input data, used to produce the results used in this paper is archived on Zenodo (https://www.doi.org/10.5281/zenodo.3782650) and has been tagged in the repository as v1.02.

*Author contributions.* A.R., J.A.S. and M.M. conceived the model design and features. A.R. wrote the model code with contributions from the remaining authors. All authors contributed to the model testing and writing the manuscript.

*Competing interests.* Heiko Goelzer is a member of the editorial board of the journal.

*Acknowledgements.* We would like to thank Mahé Perrette, Christophe Dumas, Gunter Leguy and Bill Lipscomb for valuable discussions
5   about model design that improved Yelmo, Akira Nishida for help with Lis and Ilaria Tabone and Javier Blasco for extensive model testing at intermediate development points. We are also grateful to the reviewers for helpful comments.

This research has been supported by the Spanish Ministry of Science and Innovation project RIMA (grant no. CGL2017-85975-R). Alexander Robinson was funded by the Ramón y Cajal Programme of the Spanish Ministry for Science, Innovation and Universities (grant no. RYC-2016-20587). Heiko Goelzer has received funding from the program of the Netherlands Earth System Science Centre (NESSC),
10   financially supported by the Dutch Ministry of Education, Culture and Science (OCW) under grant no. 024.002.001. Ralf Greve was supported by the Japan Society for the Promotion of Science (JSPS) KAKENHI grant nos. JP16H02224, JP17H06104 and JP17H06323, by the Japanese Ministry of Education, Culture, Sports, Science and Technology (MEXT) through the Arctic Challenge for Sustainability (ArCS) project and through the Arctic Challenge for Sustainability (ArCS) project (Program Grant Number JPMXD1300000000).

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

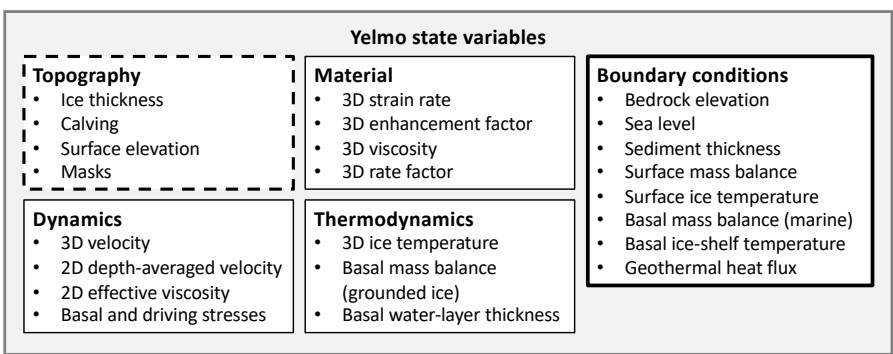

**Figure 1.** Overview of the Yelmo model structure highlighting state variables in the four components: topography, dynamics, material and thermodynamics, as well as the boundary conditions required to run the model. The thick black border for boundary variables indicates that these fields are never modified internally by Yelmo, while the components with a thin black border or dashed line are allowed to be modified depending on the context. When, for example, the topography is updated (dashed line), no other components are allowed to be modified (solid lines).

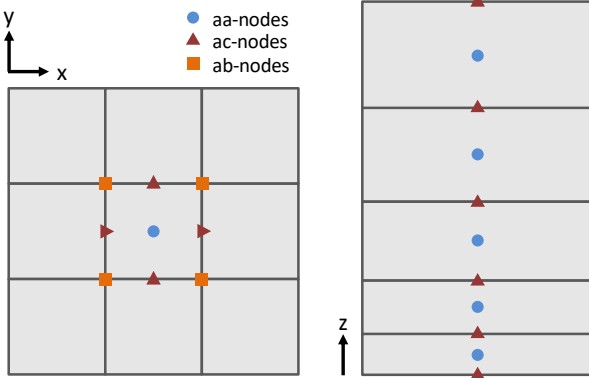

**Figure 2.** Yelmo staggered grid definition and nomenclature. The horizontal grid (left) assumes constant resolution in the x- and y-directions, while in the vertical (right) variable resolution is allowed. With any given cell defined as a 3D box, scalar variables are calculated on cell centers (aa-nodes), velocities are calculated on cell faces (ac-nodes, edges in 2D), and scalar coefficients are calculated on cell edges (ab-nodes, corners in 2D). Figure design adapted from Hoffman et al. (2018).

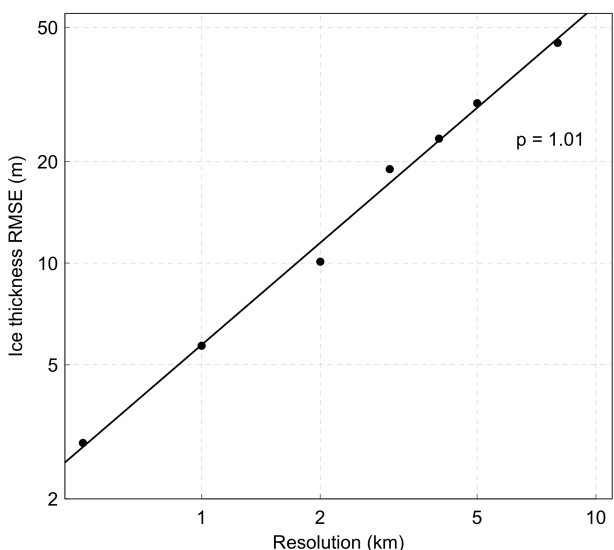

**Figure 3.** Root mean square error (RMSE) of the Halfar dome experiment after 200 yr simulated by Yelmo compared to the analytical solution versus model resolution. The value of $p = 1.01$ indicates the order of convergence as the resolution increases.

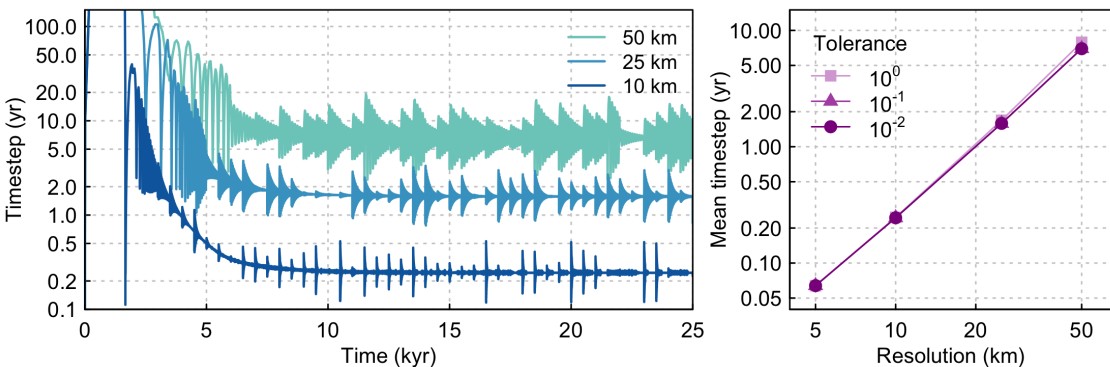

**Figure 4.** Adaptive timestepping for the EISMINT1 moving margin experiment. Time series of the timestep used by Yelmo for grid resolutions of 50 km, 25 km and 10 km and a tolerance of $\epsilon = 10^{-2}$ (left), and the mean adaptive timestep in the time range of 15-25 kyr versus model resolution (right). Separate lines in the right panel show results for different values of the tolerance parameter $\epsilon$.

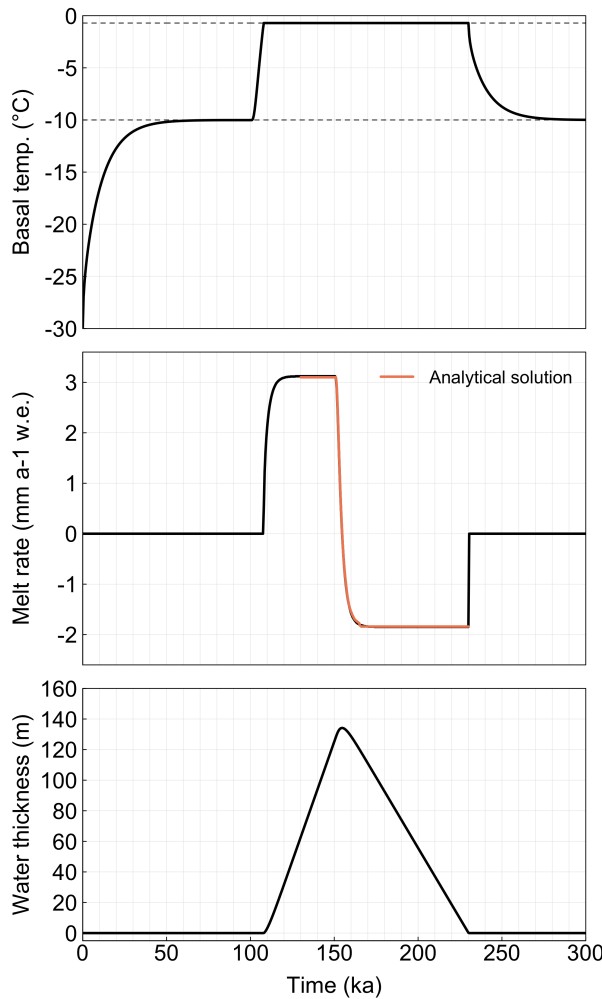

**Figure 5.** Time series of the basal temperature (top), basal melt rate (middle) and basal water layer (bottom) corresponding to the thermodynamic benchmark experiment Exp. A (Kleiner et al., 2015). The analytical solution (thick, light-red line) for the basal melt rate is compared to Yelmo results (black lines). Not that where the Yelmo results are not visible, they overlap with the analytical solution.

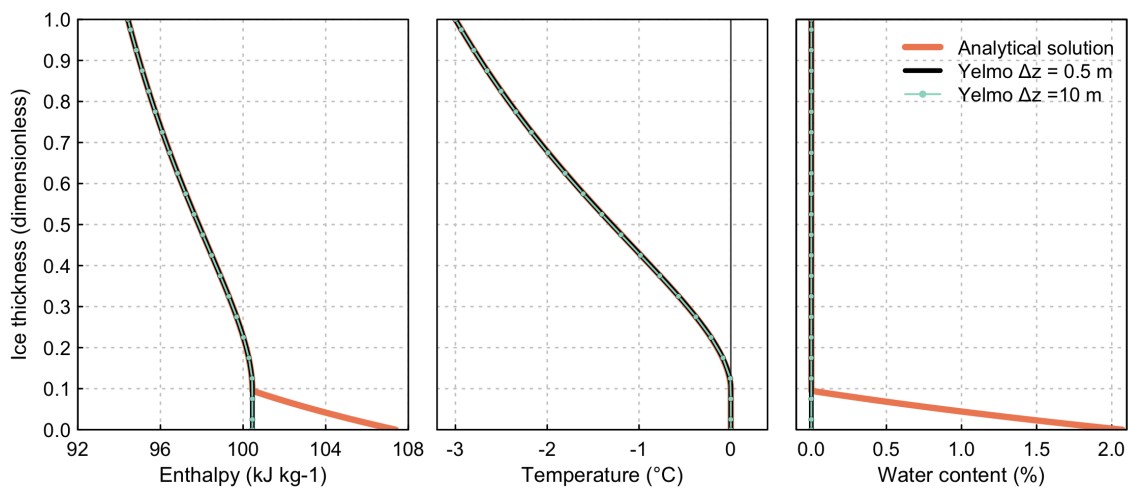

**Figure 6.** Steady-state vertical profiles of enthalpy (left), temperature (middle) and water content (right) for the thermodynamic benchmark experiment Exp. B (Kleiner et al., 2015). The analytical solution (thick, light-red lines) is compared to Yelmo results for a vertical resolution of $\Delta z = 0.5\,\text{m}$ ($n_z = 400$, black lines) and $\Delta z = 10\,\text{m}$ ($n_z = 20$, light green lines). The vertical, grey line in the middle panel shows the pressure-melting point as prescribed in this experiment. Note that where the analytical solution is not visible, it overlaps with the Yelmo results.

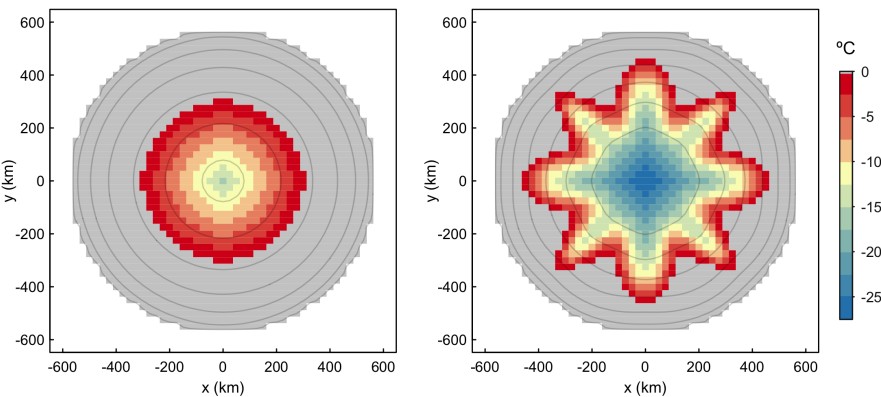

**Figure 7.** Steady-state, basal homologous temperature ($^\circ$C) distribution after 100 kyr obtained by Yelmo in EISMINT2 test A (left) and test F (right). Areas that have reached the pressure-melting point have been shaded grey. The contour lines represent ice thickness at 500 m intervals up to 3500 m.

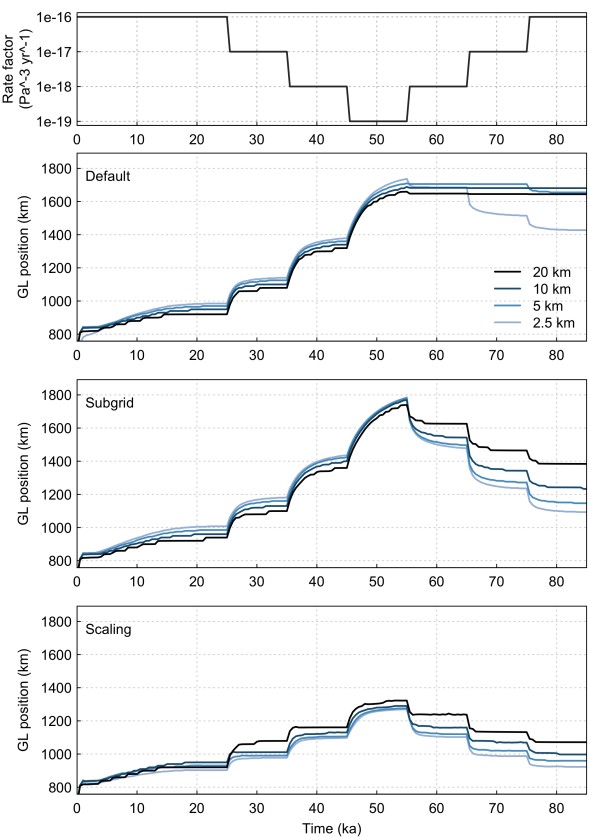

**Figure 8.** Yelmo performance in the MISMIP bedrock advance and retreat simulations on a linear sloping bed. The top panel shows the imposed rate factor $A$, with 10 kyr steps of decreasing and then increasing values. The three lower panels show the grounding line position evolution for each of three model configurations, respectively: "Default" is the standard model setup, with no special treatment of friction at or near the grounding line, "Subgrid" uses the grounded fraction at the grounding line to scale the basal friction, "Scaling" applies both the grounded fraction, and imposes a linear reduction in basal friction as the ice sheet approaches flotation. Separate simulations were run for resolutions ranging from 20 km down to 2.5 km.

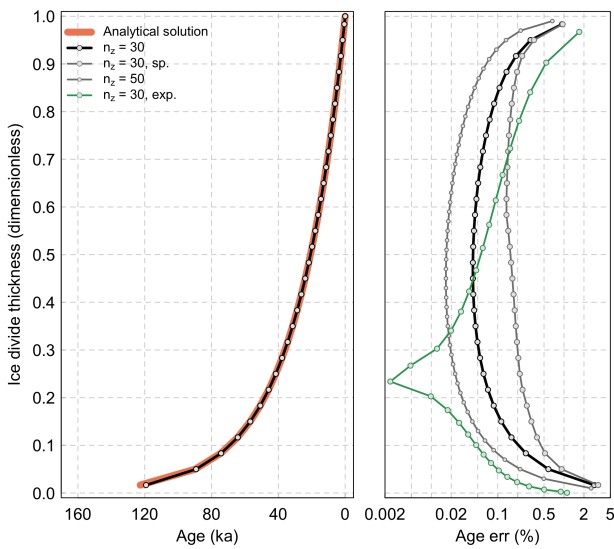

**Figure 9.** Analytical age-depth profile for idealized summit compared to Yelmo Eulerian age tracing. The ice age relative to present day (left) is shown for the analytical solution (thick, light-red line) and Yelmo with a resolution of $n_z = 30$ with a linear vertical axis and compiled at double precision (black line). The associated relative error (right) is given for this case (black line), as well as for a higher resolution of $n_z = 50$ and for $n_z = 30$ compiled at single precision (grey lines), and finally for a resolution of $n_z = 30$ compiled at double precision, but with exponentially increasing resolution at the base instead of a linear axis (green line).

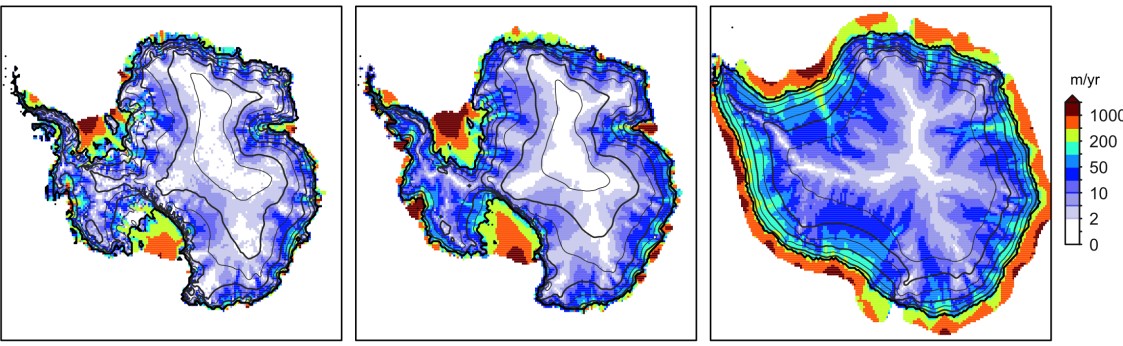

**Figure 10.** Antarctica present-day ice sheet configuration and surface velocities from observations (left), compared to a steady-state simulation with Yelmo (middle). In contrast, Antarctica glacial configuration and surface velocities simulated by Yelmo (right). Simulations were performed at 32 km resolution. The colors show surface velocity in $m\,yr^{-1}$ and the dark grey contours show surface elevation in 500 m intervals (thick lines correspond to 1000, 2000 and 3000 m above sea level). The black line shows the grounding line position.

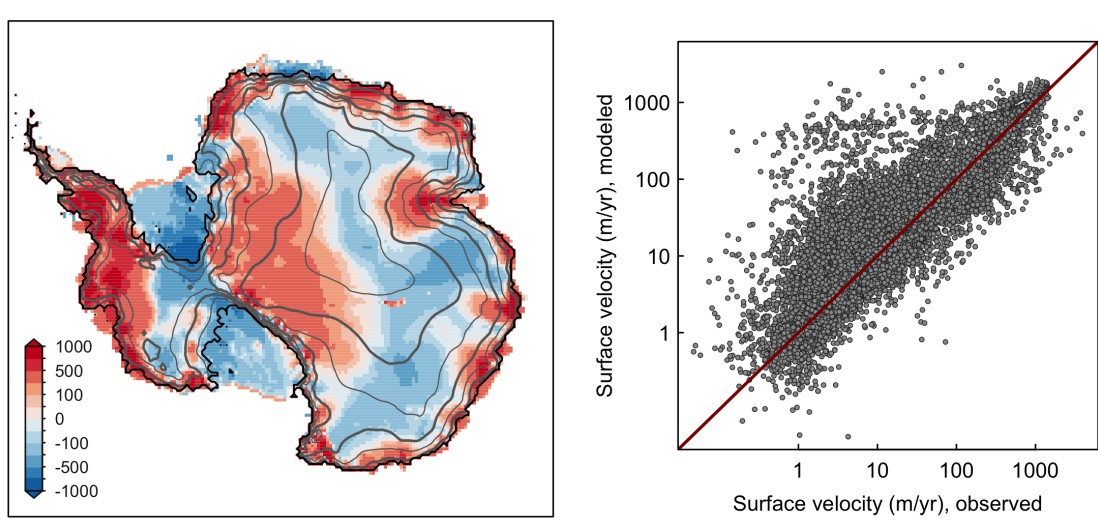

**Figure 11.** Simulated present-day ice-thickness minus observations (left) and simulated versus observed ice surface velocity (right). The colors in the left panel show the ice thickness difference in m and the dark grey contours show surface elevation in 500 m intervals (thick lines correspond to 1000, 2000 and 3000 m above sea level). The black line shows the grounding line position. In the right panel, the dark red line indicates a perfect correlation between model and observations.

**Table 1.** Yelmo performance in the EISMINT1 moving margin experiment ("Moving"), as well as in the EISMINT2 experiments A and F. Where available, metrics with the ensemble mean and standard deviation from the original benchmark experiments are also provided for comparison.

| Experiment | Model | Volume [$10^6$ km$^3$] | Area [$10^6$ km$^2$] | Melt fraction | Divide thickness [m] | Divide basal temperature [K] | Divide homologous basal temperature [$^\circ$C] |
|---|---|---|---|---|---|---|---|
| Moving | EISMINT1 | – | – | – | $2997.5 \pm 7.4$ | – | $-13.40 \pm 0.56$ |
| | Yelmo | 1.980 | 1.003 | 0.66 | 3006.6 | 257.2 | -13.37 |
| A | EISMINT2 | $2.128 \pm 0.073$ | $1.034 \pm 0.043$ | $0.72 \pm 0.15$ | $3688 \pm 48$ | $255.6 \pm 1.4$ | – |
| | Yelmo | 2.170 | 1.031 | 0.75 | 3678 | 254.7 | -15.26 |
| F | EISMINT2 | – | – | – | – | – | – |
| | Yelmo | 2.373 | 1.031 | 0.55 | 4266 | 240.7 | -28.80 |