# Peer review of "Description and validation of the ice-sheet model Yelmo (version 1.0)"

_Geoscientific Model Development, 2019_

## Short Comment (SC1) · 2 Oct 2019

Panel b of Figure 2 is very similar to panel b of Figure 1 in Hoffman, et al. (GMD, 2018, https://www.geosci-model-dev.net/11/3747/2018/gmd-11-3747-2018.html), see attached image. The authors may want to consider referencing the previous paper in the figure caption.

[Figure]

3750

Figure 1. MALI grids. (a) Horizontal grid with cell center (blue circles), edge midpoint (red triangles), and vertices (orange squares) identified for the center cell. Scalar fields ($H$, $T$) are located at

id definition and nomenclature. The horizontal grid (left) assumes constant resoluti

**Fig. 1.**

---

## Author Comment (AC1) · 3 Oct 2019

Thank you very much for the comment. Indeed Fig. 1 from Hoffman et al. (2018) inspired our own figure. We are happy to make the reference more explicit, and will do so in the revised manuscript.

―――――――――――――――――

---

## Referee Comment (RC1) · Anonymous Referee #1 · 14 Nov 2019

General comments

Robinson et al. present a new ice sheet model. Using a zero-order hybrid SIA/SSA scheme, it is computationally inexpensive compared to higher-order and full-Stokes models. The manuscript is nicely written and provides a thorough description of the physics and its implementation in the model. In addition, the Yelmo model is available on a git repository with sufficient information to run it for a few standard configurations. The paper is worth publishing although I have a few comments and suggestions that could be considered.

Main specific comments

- Most readers would probably like to see a more in-depth discussion on how the model

perform for a real ice sheet configuration. While I applaud the authors for performing the EISMINT and MISMIP experiments since they are very informative on the model behaviour, I also think that the Antarctic ice sheet experiments are a bit weak and are briefly described (e.g. length of the simulation?). First, it could be nice to have more diagnostics in addition to the sole map of surface velocities (topography error but also other fields such as basal temperature / hydrology?). It is not necessary to show a perfect match with observations (which is most of the time achievable with a dedicated tuning), but it is interesting to see the bias structure to see if it is similar to other models with a similar complexity. Second, and more importantly, it could be very nice if you could discuss transient simulations. Since the model is suited for long integrations, some glacial-interglacial simulations (even with an idealised forcing) would be very interesting. If this is not possible, alternatively, you could maybe do the InitMIP experiments, since they are relatively easy to set up, and discuss your model results with respect to what is shown in Seroussi et al. (2019)?

- Unless I am mistaken, the model does not contain an isostasy model. This is certainly a limitation and might prevent its use for glacial-interglacial applications. Do you plan to account for this in the future?

- Since you use an adaptive time step, a dedicated section could be very useful.

- Running Yelmo on my computer, I was not able to reproduce the results you show for the Antarctic ice sheet with the standard configuration file provided at the zenodo link. It is minor since I was simply digging for more info (basal drag coefficient value, trends, basal temperature etc.). Consider updating the configuration files for consistency with the results shown in the manuscript.

Other specific comments

- P11L22-25 Since beta=f(ub) for non-linear friction laws, does this mean that you have to do iterative loops (relaxation) to estimate beta for a given time step?

- P14L13-20 The reader should be reminded that these experiments are SIA only.

- P15L8-10 Why is the motivation behind this choice of model parameters. You did the Antarctic experiments with a linear friction law, it would have been more useful to use the same model configuration for the two examples shown, wouldn't it?

Technical corrections

- P2L7 Typo (millenial).

- P2L7 Not only for palaeo, future multi-millenial change of ice sheets is also of interest

- P7L9 n=3?

---

## Short Comment (SC2) · 18 Nov 2019

The paper under review is a model description of Yelmo, a new ice sheet model written in Fortran. The model is open source, and this reviewer successfully examined and downloaded the source from github, and compiled and ran the model, with unimportant technical difficulties only.

Yelmo has a very conservative design, with essentially no new physical features or submodels, at least as described here. If the publication standard at GMD is that the model is geophysical and is described by the submitted paper then this standard is unequivocably met. However, a basic description of a model, as part of its (evolving) documentation, should be part of its source code release. It is not clear that such

model documentation is a publishable document; indeed a user's or developer's manual should match the changing versions of a code and address how specific capabilities are exploited. Because I suppose that the publication standard here is beyond that of such documentation, I have the following major concerns.

Concern 1. A new model should be justified by new directions for research, new ideas, and new capabilities. The most important concern: what is the direction of this work? Substantial effort has been expended on Yelmo but it is not actually clear for what purpose. Open source ice sheet models exist with its capabilities, or with substantially greater capability (e.g. BiSCLES, CISM, ISSM, OGGM, PISM, Sicopolis), and all of these are forkable and to varying degrees modular. The authors of Yelmo chose to develope a new model, and not to add new capabilities to an existing model, despite some ideas coming from Sicopolis. So, where is this new one going?

Concern 2. The "intended for collaborative development" and "flexible and user-friendly infrastructure ..." claims in the abstract, and repeated in various ways in the paper, are not demonstrated in any substantial way. For example, there is no demonstration that only minimal code extensions are needed to add a new capability. (Presumably this would be a great deal easier in e.g. Python than in Fortran anyway.) Does the model actually represent improved infrastructure for adding new capabilities? Noting that modularity does not, by itself, imply extensibility, if the model is extensible then the paper should demonstrate it.

Concern 3. The model verification mistakes of the past are recapitulated here. The EISMINT1 moving margin (MM) "benchmark" represented a failure of the community to read the literature 25 years ago, but there is no excuse now. As clear from textbooks (van der Veen, 2013, 2nd ed.) and well-known paper papers (Bueler et al, 2005), the Halfar (1983) exact solution is a full replacement for the MM experiment, which offers exact knowledge of what an SIA model should do. Regarding the EISMINT2 and MISMIP stuff, there is some excuse for using the benchmarks (though no evidence is given that the Yelmo runs offer more than the most common capabilities). The agemodel testing via a (divide) analytic solution is applauded.

Concern 4. Does the model run in parallel? The paper does not demonstrate or consider this but the tools seem to be suitable for it. In particular the Lis linear algebra library (ssisc.org/lis/), and the biconjugate gradient method, are used for the membrane stress solver component (SSA), and the library claims parallel performance. The concern here is that a lack of parallelism is exactly one of the limitations of many previous ice sheet models, thus limiting their attainable resolution.

Now we turn from concerns to suggestions and questions.

The disadvantage of the GMD paradigm, of peer-reviewed publication of snapshot model descriptions, even users' manuals, is clear in this paper. Author and reviewer effort is devoted to a rapidly-out-of-date paper instead of (for example) devoting that effort to improving the software itself, or its evolving online documentation. Please see the mission statement of a different publication model: https://joss.theoj.org/about. The idea of the Journal of Open Source Software is to treat the software itself, and its evolving online infrastructure, as the reviewable object.

A surprise about the design of Yelmo was the decision to use only a temperature variable in the energy conservation equation. To a significant degree this means energy is not actually conserved. The alternatives, of course, include the replacement of temperature by an enthalpy variable, or the addition of a field to make a temperature/water-content pair; either makes possible energy conservation in polythermal ice. On the one hand there is no question that near-base polythermal ice is a concern for all ice sheet simulations; I would not want to see any serious treatment of ice sheet time scales without it. On the other hand, one would want the model to also be flexible enough to work for temperate mountain glaciers or Greenland outlet glaciers. The authors seem aware of this issue, but to have simply not bothered. To the extent that there is a question here it is: what was gained by this choice?

The above paragraph deliberately ignores the second-to-last sentence of the paper

reporting a "plan to transition to" enthalpy, which only begs the question. Why not build the model based on the current state of the art from the beginning? This reviewer would be delighted to see a failed-or-not attempt to build a highly-principled and highly-conserving new model instead of a recapitulation of prior deficiencies.

This reviewer had understood that Sicopolis (Greve) and GRISLI (Ritz) were models under active development which had capabilities roughly a superset of Yelmo. Is this true or not? Is this a fork of either model? (It seems not.) Is Yelmo already justified by its modularity and API design, somehow lacking in Sicopolis and/or GRISLI? It would help readers to expose these aspects of the design.

The Antarctica validation results are acceptable but suggest no capability which would draw-in researchers to use this model. To compare rather directly, a just-published paper suggests the power of many modern ice sheet models to account for the dynamics of the Antarctic ice sheet, namely the initMIP-Antarctica paper Seroussi et al 2019 (https://www.the-cryosphere.net/13/1441/2019/). It lists 16 models of Antarctica (and 33 researchers), almost all of which would seem to have capabilities equal to or exceeding that of Yelmo. So where is this model going that is different, and why should we hope for new knowledge from it?

In summary, a new model like Yelmo, containing no significant new physics or model mechanisms, could in theory be useful. It could be a better piece of software than other offerings, it might have better performance, or it might be able to process input data faster or more flexibly, or it might just be implemented better and have better documentation. This paper does not convince me of any of it.

Line-by-line, generally minor comments:

p 2 line 1: Lipscomb et al 2019 does not solve Stokes.

p 9 line 23: Hard-coding BCG for the SSA solve is a bad idea. Have there been experiments with AMG in Lis? Can a performance comparison be reported?
p 11 section 2.4: I am surprised by the temperature variable. If the claim is that converting to/from the enthalpy variable is too costly, then this should be stated.

p 14 line 13: The EISMINT1 moving margin experiment has no justification *whatsoever*. Please use the Halfar (1983) solution so that you know the exactly-correct prediction of the SIA. See "Test B" in Bueler et al (2005), and add "Test C" from that source if you want variety.

p 14 line 16: "and thermodynamic": I hope not! EISMINT1 results depend on constant temperature (isothermal) ice, so energy will not be conserved.

p 14 line 17: ""Type-I" discretization models": The fact that Yelmo agrees with other particular numerics is not relevant; EISMINT1 reported groups of results that way so as to expose a flaw not propose a standard. Instead, please take the opportunity to compare model results to exact predictions (a.k.a. analytical solutions) of the continuum model when available, *which they are in this case*. Beyond Halfar (1983) and Bueler et al (2005) for the SIA, there are exact solutions of the SSA, including results in van der Veen 1983), Schoof (2006), and Bueler (2014).

p 14 line 29: This idea of smoothing is described in the reference Bueler et al (2007)as "non-physically 'smeared'". Whatever the meaning of this fiddle, it is not physics and the reference says that. (I think the point was that if *only* the temperature variable is symmetrized then the instability goes away, which modelers probably knew at the time but none had reported concretely. That is, the instability does not occur in a variable-softness model unless the softness variation is transported in 3D.) Supposing the EISMINT2 nonsense is valuable at all, please don't offer model users this smoothing, which hides physics.
* * *

---

## Referee Comment (RC2) · Fuyuki SAITO (Referee) · 29 Nov 2019

This paper describes the numerical ice-sheet model Yelmo version1.0. The Yelmo model is available on a git repository with sufficient information to run, and this manuscript contains mostly enough description of the model physics and example application. I think this paper is fairly well written with some exception below, and can be accepted with minor revision.

(1) One point is about symmetry of the model (P14, L21 and after). Figure 3 is the simulated basal temperature of the experiments A and F of EISMINT2 configuration, and the paper states 'Yelmo produces symmetrical temperature patterns in both experiments,' First, minor one, I suggest to describe white kind of symmetry is the topic in

this section. The configuration of EISMINT2 is 'radially' symmetric, but we all failed to simulate true radially symmetric pattern in particular for experiment F.

Second, major one: what is the degree of symmetry in the argument of this section? Actually, taking a closer look, some breaks of symmetries are already visible in the figure 3. The result of experiment A looks symmetric both along X and Y axis, but not along x=y diagonal. The result of F, even worse, shows breaks of symmetry both along X and Y axis (I attached a copy of figure 3 with marks to show the breaks of symmetry). So, even under the figure resolution, Yelmo already failed to produce symmetrical temperature patterns.

In my opinion, preservation of model symmetries requires full control on the source and compilation, because even single change of arithmetic orders (e.g., (A+B)+C vs A+(B+C)) in a model can trigger and amplify breaks of symmetries. Yelmo depends on an external library in order to solve SSA equations, which is hardly controlled from outside, therefore Yelmo may find such symmetry breaking under an idealized configuration with ice shelves, even the SIA part is perfect.

On the other hand, although preservation of the symmetries in the model is desired, it is not a top priority of a model, especially for one to simulate realistic worlds. We believe that such minor points have little influence on the simulation under realistic, highly asymmetric configuration for most of the application.

So, I suggest the authors to keep the argument of Yelmo symmetries, and also state clearly the standpoint and/or main targets of Yelmo.

It may not be a reviewer's work, I check the Yelmo source code to find the source of symmetry breaking. I attached some suggestion for Yelmo to preserve the numerical symmetries as a series of patch files since revision ed94c608516e2c46c7985ea98eea94fce47b37d8 (you can run git-am to apply them). It may be not complete and, honestly speaking, it may have bugs because I did not check in detail. The author can import if they like them, but hope them to check the

revision and results in detail before inclusion. If fortunate, SIA results will be more symmetric than before.

(2) Another point is about precision (P4, L2). (See also minor points below for terminology of floating-point types).

The paper states that single and double precision give equivalent results, because the units of all time variables in Yelmo are cast in years instead of seconds and thus very small numbers are avoided. I do not understand this statement. I do not claim for the result but for the reasoning.

As far as all the quantities are larger than the smallest limit of floating-point number representation, same precision (significant digits) is kept either for the case with unit seconds and with unit years, because it differs not in precision but in the order of magnitude. The smallest number of a typical 'single' precision is around $1.18e{-}38$. What variables do have possibility to show smaller value than this? The rate factor can be small, but even ice temperature is -100 degree Celsius, its magnitude is 1e-31 /s/Pa^3, which is large enough to be represented by single precision.

I agree that, a typical number of significant decimal digits of single precision is 7, which is actually smaller than the digits of the factor from year to second (31556926, 8 digits). If all the quantities in the model is originally defined with unit year, then it is possible to meet such situation, where unit-second version shows different results in the final digit. However, many parameters are originally defined in unit second and converted into unit year in the model, thus round-off happens in some parameters themselves instead, which are almost the same situation as the unit-second case. (By the way, fortunately 31556926 can be fully represented by single floating-point number while 3155692[57] are not).

If there is a variable to be smaller than the limit in the case of unit-second, my question is solved. So, please give me an example.

Again but from a different point of view: this argument should depend on the model spatial resolution. As I mentioned, difference between single and double precision is merely the number of significant digits, if order of magnitude of all the quantities can be represented enough by the single. For a coarse resolution, difference in values at two adjacent grids of a field (e.g., surface elevation) is large enough to keep precision in their differences, However, for a higher resolution where the values at two become closer, so-called cancellation effects become large enough to reduce the precision of their difference. Relative error of the difference can be large enough for single precision to deviate from that by double precision. Generally speaking, a higher resolution experiment require high-precision computation to avoid such cancellation effects.

Minor points:

About precision (P4, L2) 'Single' and 'double' precision are in principle machine dependent characteristics although there are few exceptions. There is some definition of typical floating-point representation in IEEE754.

2.3, around Eq.(14). ub is defined as basal sliding above Eq.(14) while a depth-averaged velocity below (14). I am confused. Possibly typo?

2.4, below Eq. (27). Better to write as 'Horizontal diffusion is assumed negligible.'

Table 1 last column. No degree mark.

SAITO Fuyuki.

Please also note the supplement to this comment:
https://www.geosci-model-dev-discuss.net/gmd-2019-273/gmd-2019-273-RC2-supplement.zip

[Figure]

**Fig. 1.** Annotation on Figure 3 in the paper.

---

## Referee Comment (RC3) · Fuyuki SAITO (Referee) · 2 Dec 2019

There are two additional things to my first review.

First one, minor. There are contour lines in the figure 3, but what are they? I suppose they are surface elevation field. Anyway please explicitly describe them. I am sorry not to point out this in the first report. I just realized this now.

Second one, also minor. I prepare two more patches in order to preserve symmetries. I confirmed symmetries of basal temperature field in X-axis Y-axis and diagonals at least for experiment F which was executed on my PC (compilation is by gfortran). Attached is a series of patches from revision ed94c608516e2c46c7985ea98eea94fce47b37d8, which includes all the patches I previously posted, so you can discard the previous one.

[Figure]

SAITO Fuyuki.

Please also note the supplement to this comment:
https://www.geosci-model-dev-discuss.net/gmd-2019-273/gmd-2019-273-RC3-supplement.zip

———————————————————

---

## Author Response (AR1)

To the editor:

Please find our point-by-point responses to the reviewer comments below. We have addressed all major comments in the updated manuscript. A latex-diff generated file is also included at the end of this document. Unfortunately, the resulting document produced some errors and does not highlight the changes very well. Therefore, we list the major changes to the manuscript below for your reference:

- Following the suggestion of Reviewer 1, we added a new section describing the timestepping methodology of the model (Section 4). We have also included a new figure demonstrating the capability of the adaptive timestepping scheme (Figure 4), which is discussed in the model validation section.
- To address Reviewer 1's comments on the Antarctica simulations, more information has been provided on the simluations in the Antarctica section, including a new figure showing the error in ice thickness and a scatterplot of modeled versus observed surface velocity (Figure 11).
- Following the suggestion of Ed Bueler, we have included a new figure showing Yelmo's performance for the Halfar dome benchmark test (Figure 3), which is discussed in the model validation section.
- To address Ed Bueler's concerns regarding the thermodynamics solver, we have included two new figures that show Yelmo's performance for the enthalpy benchmark experiments of Kleiner et al. (2015) (Figs. 5 & 6), which are discussed in the model validation section.
- Since first submitting the manuscript, we have included the local evolution of the basal water layer as a prognostic variable in the model, including a new equation (Eq. 29).

These changes, as well as all additional modifications, are discussed below. Thank you for your consideration.

========== Short comment 1 (Matthew Hoffman) ====================

Panel b of Figure 2 is very similar to panel b of Figure 1 in Hoffman, et al. (GMD, 2018, https://www.geosci-model-dev.net/11/3747/2018/gmd-11-3747-2018.html), see attached image. The authors may want to consider referencing the previous paper in the figure caption.

Thank you very much for the comment. Indeed Fig. 1 from Hoffman et al. (2018) inspired our own figure. We are happy to make the reference more explicit, and have done so in the caption of the figure revised manuscript.

========== Reviewer comment 1 (anonymous) =======================

We thank the reviewer for bringing up several key points that will serve to improve the manuscript. We have addressed these concerns below.

General comments

Robinson et al. present a new ice sheet model. Using a zero-order hybrid SIA/SSA scheme, it is computationally inexpensive compared to higher-order and full-Stokes models. The manuscript is nicely written and provides a thorough description of the physics and its implementation in the model. In addition, the Yelmo model is available on a git repository with sufficient information to run it for a few standard configurations. The paper is worth publishing although I have a few comments and suggestions that could be considered.

Thank you for the very positive evaluation.

Main specific comments

- Most readers would probably like to see a more in-depth discussion on how the model perform for a real ice sheet configuration. While I applaud the authors for performing the EISMINT and MISMIP experiments since they are very informative on the model behaviour, I also think that the Antarctic ice sheet experiments are a bit weak and are briefly described (e.g. length of the simulation?). First, it could be nice to have more diagnostics in addition to the sole map of surface velocities (topography error but also other fields such as basal temperature / hydrology?). It is not necessary to show a perfect match with observations (which is most of the time achievable with a dedicated tuning), but it is interesting to see the bias structure to see if it is similar to other models with a similar complexity. Second, and more importantly, it could be very nice if you could discuss transient simulations. Since the model is suited for long integrations, some glacial-interglacial simulations (even with an idealised forcing) would be very interesting. If this is not possible, alternatively, you could maybe do the InitMIP experiments, since they are relatively easy to set up, and discuss your model results with respect to what is shown in Seroussi et al. (2019)?

We believe that more complex simulations fall outside the scope of this manuscript, as we would like to limit this model description paper to the ice-sheet model itself, and we expect more realistic simulations to be forthcoming soon in future studies. But we agree with the reviewer that more information about the Antarctica simulations could be provided. The present-day simulation is indeed comparable to the InitMIP experiment and the relevant metrics (rmse[H], rmse[vel], rmse[logvel]) already appear in the text. Nonetheless, we have added an additional plot explicitly showing the ice thickness bias and comparison of modeled velocity with observations, with relevant discussion.

- Unless I am mistaken, the model does not contain an isostasy model. This is certainly a limitation and might prevent its use for glacial-interglacial applications. Do you plan to account for this in the future?

Yelmo itself was designed specifically as a modular ice-sheet model (ice flow and thermodynamics). Consequently, it does not include an isostasy model, and we believe this is actually one of its important characteristics. Classically, ice-sheet models have been packaged as full system tools that contain several components performing isostatic, surface mass balance and shelf-melt calculations, among other things. Here we would rather like to publish an ice-sheet model in its minimal form. Numerous GIA models are available that can easily be coupled to Yelmo.

> - Since you use an adaptive time step, a dedicated section could be very useful.

This is a good point. We have added a section to discuss the time stepping method and have added an additional figure for demonstration.

> - Running Yelmo on my computer, I was not able to reproduce the results you show for the Antarctic ice sheet with the standard configuration file provided at the zenodo link. It is minor since I was simply digging for more info (basal drag coefficient value, trends, basal temperature etc.). Consider updating the configuration files for consistency with the results shown in the manuscript.

Thank you, this feedback is appreciated. The code archived at Zenodo for the published paper has been confirmed to work in all documented cases.

> Other specific comments
>
> - P11L22-25 Since beta=f(ub) for non-linear friction laws, does this mean that you have to do iterative loops (relaxation) to estimate beta for a given time step?

Yes, we iterate over beta and viscosity in a typical way. This is discussed in detail for beta when the SSA equations are introduced (P10L1).

> - P14L13-20 The reader should be reminded that these experiments are SIA only.

Yes, this has been added.

> - P15L8-10 Why is the motivation behind this choice of model parameters. You did the Antarctic experiments with a linear friction law, it would have been more useful to use the same model configuration for the two examples shown, wouldn't it?

The MISMIP benchmark experiments used these basal friction parameter values, so they are used here. In the Antarctica experiments, we wanted to keep the setup simple, so it was unnecessary to prescribe the particular values from MISMIP. We hope, as well, that it shows that it is easy to change from one law and parameter values to another.

Technical corrections

- P2L7 Typo (millenial).

Ok, this has been fixed.

- P2L7 Not only for palaeo, future multi-millenial change of ice sheets is also of interest

Yes, indeed, this has been changed.

- P7L9 n=3?

Correct, this has been fixed.

========== **Short comment 2 (Ed Bueler)** ============================

We are grateful to Ed Bueler for providing critical comments regarding our work.  We have considered all comments, and provide a point-by-point response below.

The paper under review is a model description of Yelmo, a new ice sheet model written in Fortran. The model is open source, and this reviewer successfully examined and downloaded the source from github, and compiled and ran the model, with unimportant technical difficulties only.

Yelmo has a very conservative design, with essentially no new physical features or submodels, at least as described here. If the publication standard at GMD is that the model is geophysical and is described by the submitted paper then this standard is unequivocally met. However, a basic description of a model, as part of its (evolving) documentation, should be part of its source code release. It is not clear that such model documentation is a publishable document; indeed a user's or developer's manual should match the changing versions of a code and address how specific capabilities are exploited.

It is true that the physics contained within Yelmo are rather conservative. However, the model design is rather unique, compared to many other models, and we believe this to be an added value to the community. A large section of the manuscript has been dedicated to explaining the model design from a technical point of view (not just numerics), which we believe is an important part of any model description paper and is often overlooked.

Because I suppose that the publication standard here is beyond that of such documentation, I have the following major concerns.

Concern 1. A new model should be justified by new directions for research, new ideas, and new capabilities. The most important concern: what is the

direction of this work? Substantial effort has been expended on Yelmo but it is not actually clear for what purpose. Open source ice sheet models exist with its capabilities, or with substantially greater capability (e.g. BiSCLES, CISM, ISSM, OGGM, PISM, Sicopolis), and all of these are forkable and to varying degrees modular. The authors of Yelmo chose to develop a new model, and not to add new capabilities to an existing model, despite some ideas coming from Sicopolis. So, where is this new one going?

One key phrase in this comment is "to varying degrees modular". It is not just modularity that is important (which is very important!), but also encapsulation. We built Yelmo explicitly to have a simple API that is intuitive and encapsulated (requires minimum definition of objects in external programs). Most notably, Yelmo should be quite capable for coupling with intermediate complexity climate models. We have tried to explain this motivation in the Introduction and Model design sections, but this has been further clarified in the revised manuscript. Nonetheless, we would point out that Yelmo has already proven extremely useful to us and we have noted interest from others, which indicates the demand for such a model, despite alternatives being available.

> Concern 2. The "intended for collaborative development" and "flexible and user-friendly infrastructure ..." claims in the abstract, and repeated in various ways in the paper, are not demonstrated in any substantial way. For example, there is no demonstration that only minimal code extensions are needed to add a new capability. (Presumably this would be a great deal easier in e.g. Python than in Fortran anyway.) Does the model actually represent improved infrastructure for adding new capabilities? Noting that modularity does not, by itself, imply extensibility, if the model is extensible then the paper should demonstrate it.

This is a good point. However, it was difficult to find a simple example that wouldn't overly complicate the text. The proof of these statements, we believe, is nonetheless contained in the available source code.

> Concern 3. The model verification mistakes of the past are recapitulated here. The EISMINT1 moving margin (MM) "benchmark" represented a failure of the community to read the literature 25 years ago, but there is no excuse now. As clear from textbooks (van der Veen, 2013, 2nd ed.) and well-known paper papers (Bueler et al, 2005), the Halfar (1983) exact solution is a full replacement for the MM experiment, which offers exact knowledge of what an SIA model should do. Regarding the EISMINT2 and MISMIP stuff, there is some excuse for using the benchmarks (though no evidence is given that the Yelmo runs offer more than the most common capabilities). The age- model testing via a (divide) analytic solution is applauded.

We agree that the EISMINT1/2 benchmarks are not true verification tests. However, they do serve as historical benchmarks and we believe can provide perspective on the performance of the model. Nonetheless, we have also implemented the Halfar/Bueler

exact test cases. Verification Test B in the Halfar configuration has been added in the revised manuscript.

> Concern 4. Does the model run in parallel? The paper does not demonstrate or consider this but the tools seem to be suitable for it. In particular the Lis linear algebra library (ssisc.org/lis/), and the biconjugate gradient method, are used for the membrane stress solver component (SSA), and the library claims parallel performance. The concern here is that a lack of parallelism is exactly one of the limitations of many previous ice sheet models, thus limiting their attainable resolution.

The model has been built with parallelization in mind, however its capabilities have not been tested or fully implemented. The Lis library already supports parallel solving. The thermodynamics/enthalpy solver has been designed to be solved column-by-column, facilitating parallelization (as mentioned in the text). Additional smaller routines will also be easy to parallelize. Thus, this will not be a limitation of the model, but at this stage, it was not the first priority.

> Now we turn from concerns to suggestions and questions.

> The disadvantage of the GMD paradigm, of peer-reviewed publication of snapshot model descriptions, even users' manuals, is clear in this paper. Author and reviewer effort is devoted to a rapidly-out-of-date paper instead of (for example) devoting that effort to improving the software itself, or its evolving online documentation. Please see the mission statement of a different publication model: https://joss.theoj.org/about. The idea of the Journal of Open Source Software is to treat the software itself, and its evolving online infrastructure, as the reviewable object.

The raised discussion on publication paradigms is interesting and may be taken up at a different level. However, it seems rather outside the scope of this review to debate whether the GMD paradigm is better or worse than another journal. Having submitted to GMD, we do believe there is great value in a static and citable model description, which can serve as a common reference point for future work.

> A surprise about the design of Yelmo was the decision to use only a temperature variable in the energy conservation equation. To a significant degree this means energy is not actually conserved. The alternatives, of course, include the replacement of temperature by an enthalpy variable, or the addition of a field to make a temperature/water-content pair; either makes possible energy conservation in polythermal ice. On the one hand there is no question that near-base polythermal ice is a concern for all ice sheet simulations; I would not want to see any serious treatment of ice sheet time scales without it. On the other hand, one would want the model to also be flexible enough to work for temperate mountain glaciers or Greenland outlet glaciers. The authors seem aware of this issue, but to have simply not

bothered. To the extent that there is a question here it is: what was gained by this choice?

Several models still choose to use temperature as the prognostic variable (CISM, GRISLI, IMAU-ICE, among others). Despite the lack of full energy conservation when water is present, this may not be a first-order issue for lower-resolution, long timescale simulations, given a number of other large uncertainties. In addition, the impact of water content on the rate factor is deeply uncertain. While an enthalpy solver would add a further degree of energy conservation, accurately mapping the transition from temperate to cold ice can only be achieved with very high vertical resolution, or potentially a two-layer scheme as in SICOPOLIS. In either case, this results in significant computational overhead and potentially numerical artifacts. We felt it was important now to have a fast, capable model that can run ensembles of long timescale simulations. In the revised manuscript we have added discussion of the energy scheme, and also show to what extent the temperature scheme can pass enthalpy benchmarks (Kleiner et al., 2015).

> The above paragraph deliberately ignores the second-to-last sentence of the paper reporting a "plan to transition to" enthalpy, which only begs the question. Why not build the model based on the current state of the art from the beginning? This reviewer would be delighted to see a failed-or-not attempt to build a highly-principled and highly- conserving new model instead of a recapitulation of prior deficiencies.

See above. This is mainly a question of resources during the model development phase. As stated above, incorporating an enthalpy solver only brings added value when the transition from temperate to cold ice is well resolved. To do so in a computationally efficient manner requires careful consideration of the numerical treatment, which is reserved for future work. This is now discussed in the revised manuscript.

> This reviewer had understood that Sicopolis (Greve) and GRISLI (Ritz) were models under active development which had capabilities roughly a superset of Yelmo. Is this true or not? Is this a fork of either model? (It seems not.) Is Yelmo already justified by its modularity and API design, somehow lacking in Sicopolis and/or GRISLI? It would help readers to expose these aspects of the design.

Both SICOPOLIS and GRISLI have active user bases and are undergoing development. The capabilities with respect to physics in all three models are rather comparable, but the model designs are quite different. We have extended the discussion of the model design choices made here that differentiate Yelmo from the others.

> The Antarctica validation results are acceptable but suggest no capability which would draw-in researchers to use this model. To compare rather directly, a just-published paper suggests the power of many modern ice sheet models to account for the dynamics of the Antarctic ice sheet, namely the initMIP-Antarctica paper Seroussi et al 2019 (httpss://www.thecryosphere.net/13/1441/2019/). It lists 16 models of Antarctica (and 33 researchers), almost all of which would seem to have capabilities equal to or exceeding that of Yelmo. So where is this model going that is different, and why should we hope for new knowledge from it?

The goal of the Antarctica validation is simply to show that Yelmo can produce reasonable results for a realistic domain and for very different boundary conditions. Our results are in line with the models that appear in the initMIP-Antarctica study. Despite its similar physics to other models, it is sure that were Yelmo to be included in the ensemble, its results would not be identical. So there is potentially value in contributing to model diversity. However, Yelmo's main value at the current release is in its usability and transparency, as discussed above.

In summary, a new model like Yelmo, containing no significant new physics or model mechanisms, could in theory be useful. It could be a better piece of software than other offerings, it might have better performance, or it might be able to process input data faster or more flexibly, or it might just be implemented better and have better documentation. This paper does not convince me of any of it.

This conclusion is arguably subjective, and Yelmo will not be useful for everyone. However, we have made a concerted effort in the revised manuscript to make the advantages of Yelmo clearer.

Line-by-line, generally minor comments:

p 2 line 1: Lipscomb et al 2019 does not solve Stokes.

The phrasing here is ambiguous, and now has been corrected.

p 9 line 23: Hard-coding BCG for the SSA solve is a bad idea. Have there been experiments with AMG in Lis? Can a performance comparison be reported?

We have not performed additional sensitivity tests with variations on the Lis solver options. Through experience with SICOPOLIS, the BCG solver was recommended to balance speed and accuracy. We take the reviewer's point that the settings should not be hard coded, and the option has been added to the model parameters.

p 11 section 2.4: I am surprised by the temperature variable. If the claim is that converting to/from the enthalpy variable is too costly, then this should be stated.

We do not make that claim here. See above comments on enthalpy.

p 14 line 13: The EISMINT1 moving margin experiment has no justification *whatsoever*. Please use the Halfar (1983) solution so that you know the

exactly-correct prediction of the SIA. See "Test B" in Bueler et al (2005), and add "Test C" from that source if you want variety.

While we understand the reviewer's point, one justification we see of EISMINT1 experiments is that they serve as a historical reference. As mentioned above we have now added results from Test B to the revised manuscript.

p 14 line 16: "and thermodynamic": I hope not! EISMINT1 results depend on constant temperature (isothermal) ice, so energy will not be conserved.

The phrasing has been modified. While the feedback between temperature and dynamics was disabled in the EISMINT1 experiments, the temperature field was nonetheless diagnosed and can be used for comparison.

p 14 line 17: ""Type-I" discretization models": The fact that Yelmo agrees with other particular numerics is not relevant; EISMINT1 reported groups of results that way so as to expose a flaw not propose a standard. Instead, please take the opportunity to compare model results to exact predictions (a.k.a. analytical solutions) of the continuum model when available, *which they are in this case*. Beyond Halfar (1983) and Bueler et al (2005) for the SIA, there are exact solutions of the SSA, including results in van der Veen 1983), Schoof (2006), and Bueler (2014).

When comparing with EISMINT1 reported results, it is important to distinguish the numerical discretization, since as mentioned by the reviewer, there is no analytical solution to compare with. Again, we have added a comparison with Test B to compare with an analytical result.

p 14 line 29: This idea of smoothing is described in the reference Bueler et al (2007)as "non-physically 'smeared'". Whatever the meaning of this fiddle, it is not physics and the reference says that. (I think the point was that if *only* the temperature variable is symmetrized then the instability goes away, which modelers probably knew at the time but none had reported concretely. That is, the instability does not occur in a variable-softness model unless the softness variation is transported in 3D.) Supposing the EISMINT2 nonsense is valuable at all, please don't offer model users this smoothing, which hides physics.

We have removed this comment.

========== **Reviewer comments 2 & 3 (Fuyuki Saito)** ================

We thank Fuyuki Saito for the careful review and interesting code suggestions. Please find our response below.

This paper describes the numerical ice-sheet model Yelmo version1.0. The Yelmo model is available on a git repository with sufficient information to run, and this manuscript contains mostly enough description of the model physics

and example application. I think this paper is fairly well written with some exception below, and can be accepted with minor revision.

(1) One point is about symmetry of the model (P14, L21 and after). Figure 3 is the simulated basal temperature of the experiments A and F of EISMINT2 configuration, and the paper states 'Yelmo produces symmetrical temperature patterns in both experiments,' First, minor one, I suggest to describe white kind of symmetry is the topic in this section. The configuration of EISMINT2 is 'radially' symmetric, but we all failed to simulate true radially symmetric pattern in particular for experiment F.

**This is a good point. We are now more precise when discussing symmetry with respect to these tests.**

Second, major one: what is the degree of symmetry in the argument of this section? Actually, taking a closer look, some breaks of symmetries are already visible in the figure 3. The result of experiment A looks symmetric both along X and Y axis, but not along x=y diagonal. The result of F, even worse, shows breaks of symmetry both along X and Y axis (I attached a copy of figure 3 with marks to show the breaks of symmetry). So, even under the figure resolution, Yelmo already failed to produce symmetrical temperature patterns.
In my opinion, preservation of model symmetries requires full control on the source and compilation, because even single change of arithmetic orders (e.g., (A+B)+C vs A+(B+C)) in a model can trigger and amplify breaks of symmetries. Yelmo depends on an external library in order to solve SSA equations, which is hardly controlled from outside, therefore Yelmo may find such symmetry breaking under an idealized configuration with ice shelves, even the SIA part is perfect. On the other hand, although preservation of the symmetries in the model is desired, it is not a top priority of a model, especially for one to simulate realistic worlds. We believe that such minor points have little influence on the simulation under realistic, highly asymmetric configuration for most of the application. So, I suggest the authors to keep the argument of Yelmo symmetries, and also state clearly the standpoint and/or main targets of Yelmo.

**It is an important validation of the model to show that on a domain like that of EISMINT1/2, numerical artifacts do not arise leading to blatant asymmetry in the result. However, in any realistic cases, topographic and boundary forcing will likely dominate over minor asymmetries. So, it is indeed not a critical priority, given that Yelmo performs well in this regard. This has been clarified in the revised work.**

It may not be a reviewer's work, I check the Yelmo source code to find the source of symmetry breaking. I attached some suggestion for Yelmo to preserve the numerical symmetries as a series of patch files since revision ed94c608516e2c46c7985ea98eea94fce47b37d8 (you can run git-am to apply them). It may be not complete and, honestly speaking, it may have bugs because I did not check in detail. The author can import if they like them, but

hope them to check the revision and results in detail before inclusion. If fortunate, SIA results will be more symmetric than before.

We kindly thank the reviewer for the extra effort of patching the code with these fixes. Independently, we also realized that the time stepping scheme with respect to the different components needed improvement for stability (moving from a simple Euler step to a predictor-corrector approach). This also resulted in significant improvement in the symmetry of the test. While applying your patches did improve the symmetry compared to the original model, in the updated version, the patches made no changes to an already satisfying result (see figure below, which shows the basal temperature field compared with itself mirrored along the y-axis; purely zero-error are white points).

[Figure]

(2) Another point is about precision (P4, L2). (See also minor points below for terminology of floating-point types).
The paper states that single and double precision give equivalent results, because the units of all time variables in Yelmo are cast in years instead of seconds and thus very small numbers are avoided. I do not understand this statement. I do not claim for the result but for the reasoning.
As far as all the quantities are larger than the smallest limit of floating-point number representation, same precision (significant digits) is kept either for the case with unit seconds and with unit years, because it differs not in precision but in the order of magnitude. The smallest number of a typical 'single' precision is around 1.18e−38. What variables do have possibility to show smaller value than this? The rate factor can be small, but even ice temperature is -100 degree Celsius, its magnitude is 1e-31 /s/Paˆ3, which is large enough to be represented by single precision.

I agree that, a typical number of significant decimal digits of single precision is 7, which is actually smaller than the digits of the factor from year to second (31556926, 8 digits). If all the quantities in the model is originally defined with unit year, then it is possible to meet such situation, where unit-second version shows different results in the final digit. However, many parameters are originally defined in unit second and converted into unit year in the model,

thus round-off happens in some parameters themselves instead, which are almost the same situation as the unit-second case. (By the way, fortunately 31556926 can be fully represented by single floating-point number while 3155692[57] are not).
If there is a variable to be smaller than the limit in the case of unit-second, my question is solved. So, please give me an example.

Again but from a different point of view: this argument should depend on the model spatial resolution. As I mentioned, difference between single and double precision is merely the number of significant digits, if order of magnitude of all the quantities can be represented enough by the single. For a coarse resolution, difference in values at two adjacent grids of a field (e.g., surface elevation) is large enough to keep precision in their differences, However, for a higher resolution where the values at two become closer, so-called cancellation effects become large enough to reduce the precision of their difference. Relative error of the difference can be large enough for single precision to deviate from that by double precision. Generally speaking, a higher resolution experiment require high-precision computation to avoid such cancellation effects.

The concern here is primarily with avoiding extreme value divisions (large divided by small or small divided by large numbers), as well as so-called cancelation effects. This is particularly a concern in ice sheet modeling, where velocities, basal friction coefficients, effective pressure and rate factors can vary over several orders of magnitude. The statement in the manuscript was informed by experience. However, it is not critical to the manuscript, and so has been removed.

Minor points:

About precision (P4, L2) 'Single' and 'double' precision are in principle machine dependent characteristics although there are few exceptions. There is some definition of typical floating-point representation in IEEE754.

We note that in this sentence we equate single precision with 32 bits and double precision with 64 bits. This appears to be consistent with the definition in IEEE754, so we have left it as it is.

2.3, around Eq.(14). ub is defined as basal sliding above Eq.(14) while a depth-averaged velocity below (14). I am confused. Possibly typo?

In regions of plug flow that treated by the SSA solution, basal sliding and depth-averaged velocity should be identical.

2.4, below Eq. (27). Better to write as 'Horizontal diffusion is assumed negligible.'

Ok, this has been changed.

Table 1 last column. No degree mark.

Ok, this has been added.

There are two additional things to my first review.

First one, minor. There are contour lines in the figure 3, but what are they? I suppose they are surface elevation field. Anyway please explicitly describe them. I am sorry not to point out this in the first report. I just realized this now.

These are indeed contour lines. This will be added to the caption.

[Reviewer comment 3] Second one, also minor. I prepare two more patches in order to preserve symmetries. I confirmed symmetries of basal temperature field in X-axis Y-axis and diagonals at least for experiment F which was executed on my PC (compilation is by gfortran). Attached is a series of patches from revision ed94c608516e2c46c7985ea98eea94fce47b37d8, which includes all the patches I previously posted, so you can discard the previous one.

Thank you again. Please see the comments on symmetry above.

[revised manuscript text omitted]
_{\text{g}} = H - \frac{\rho_{\text{sw}}}{\rho} \left( \max \left( z_{\text{sl}} - z_{\text{b}}, 0 \right) \right) \tag{4}$$

where $\rho$ is the ice density and $\rho_{\text{sw}}$ the seawater density, and $z_{\text{sl}}$ and $z_{\text{b}}$ are the boundary fields of sea level and bedrock elevation, respectively. $H_{\text{g}}$ can thus be positive, zero or negative. When $H_{\text{g}}$ is positive, the ice thickness exceeds the flotation criterion, and is considered grounded, while when $H_{\text{g}}$ is zero or negative, the ice is considered floating.

Yelmo also calculates the grounded fraction of each grid point, $f_{\text{g}}$. On aa-nodes, $f_{\text{g}}$ is only assigned binary values to maintain consistency with the overall grid definition: zero when $H_{\text{g}} \leq 0$ or one when $H_{\text{g}} > 0$. However, on  ac-nodes, the values of $f_{\text{g,acx}}$ and $f_{\text{g,acy}}$ are determined by linearly interpolating $H_{\text{g}}$ from the two bounding aa-nodes. When both bounding aa-nodes are positive $f_{\text{g,ac}} = 1$, and when both are negative $f_{\text{g,ac}} = 0$. When one aa-node is positive ($\sout{H_{\text{g,pos}}}H_{\text{g+}}$) and one aa-node is negative ($\sout{H_{\text{g,neg}}}H_{\text{
[revised manuscript text omitted]

(16)

where $(\tau_{b,x}, \tau_{b,y}) = -\beta(u_b, v_b)$ (or in vector notation $\tau_b = -\beta\mathbf{u}_b$) is the basal stress due to friction. The basal friction coefficient $\beta$ is set to zero for floating ice shelves, and can otherwise be set to a constant value or follow another user-defined formulation (power law, regularized Coulomb, etc.), depending on the context  (see basal friction description below for details). The depth-integrated (2D) effective viscosity, which is only used for solving the SSA dynamics, is defined as

$$\eta_d = \left[ \frac{1}{2}\left( \bar{A}^{-1/n} \right) \left( \dot{\varepsilon}_d^2 + \dot{\varepsilon}_0^2 \right)^{\frac{1-n}{2n}} \right] H$$

(17)

where $\bar{A}$ is the vertically-averaged rate factor, $\dot{\varepsilon}_d$ is the 2D effective strain rate and $\dot{\varepsilon}_0^2$ is a small regularization factor for avoiding a potential singularity when velocity gradients are zero. The 2D effective strain rate is calculated as a reduced form of the second invariant of the strain rate tensor (Eq. 9) that does not include vertical shear terms:

$$\dot{\varepsilon}_d^2 = \left( \frac{\partial u_b}{\partial x} \right)^2 + \left( \frac{\partial v_b}{\partial y} \right)^2 + \frac{\partial u_b}{\partial x}\frac{\partial v_b}{\partial y} + \frac{1}{4}\left( \frac{\partial u_b}{\partial y} + \frac{\partial v_b}{\partial x} \right)^2.$$

(18)

In Yelmo, $\dot{\varepsilon}_d$ is only used for calculating $\eta_d$, while the 3D effective strain rate is calculated from the full strain rate tensor in the material component (see Material section above). Calculating the full tensor during the iterative SSA solution procedure would

be much more computationally expensive, while the 2D effective strain rate is already sufficient for the vertically integrated

5  SSA equations (Pollard and DeConto, 2012).

The stress boundary condition imposed at the floating ice front, following Winkelmann et al. (2011) and Greve and Blatter (2009), is

$$\eta_d \left( 4\frac{\partial u}{\partial x} + 2\frac{\partial v}{\partial y} \right) n_x + \eta_d \left( \frac{\partial u}{\partial y} + \frac{\partial v}{\partial x} \right) n_y = \left( \frac{1}{2}\rho g H^2 - \frac{1}{2}\rho_{\text{sw}} g H_{\text{o}}^2 \right) n_x$$
$$\eta_d \left( 4\frac{\partial v}{\partial y} + 2\frac{\partial u}{\partial x} \right) n_y + \eta_d \left( \frac{\partial v}{\partial x} + \frac{\partial u}{\partial y} \right) n_x = \left( \frac{1}{2}\rho g H^2 - \frac{1}{2}\rho_{\text{sw}} g H_{\text{o}}^2 \right) n_y. \tag{19}$$

The depth of the seawater up to the flotation depth, $H_{\text{o}}$, is defined as: $H_{\text{o}} = \min\left( z_{\text{sl}} - z_b, \frac{\rho}{\rho_{\text{sw}}} H \right)$ $H_{\text{o}} = \min\left( \max(z_{\text{sl}} - z_b, 0), \frac{\rho}{\rho_{\text{
[revised manuscript text omitted]

|  | |  2.373 |  1.031 | |  4266 |  240.7 |  -28.80 |